# The evolution and dynamics of the Hunga Tonga-Hunga Ha'apai sulphate aerosol plume in the stratosphere

Bernard Legras[1], Clair Duchamp[1], Pasquale Sellitto[2,3], Aurélien Podglajen[1], Elisa Carboni[4], Richard Siddans[4], Jens-Uwe Grooß[5], Sergey Khaykin[6], and Felix Ploeger[5]

[1]Laboratoire de Météorologie Dynamique (LMD-IPSL), UMR CNRS 8539, ENS-PSL, École Polytechnique, Sorbonne Université, Institut Pierre Simon Laplace, Paris, France
[2]Univ. Paris Est Créteil and Université de Paris Cité, CNRS, Laboratoire Interuniversitaire des Systèmes Atmosphériques (LISA-IPSL), Institut Pierre-Simon Laplace, Créteil, France
[3]Istituto Nazionale di Geofisica e Vulcanologia (INGV), Osservatorio Etneo (OE), Catania, Italy
[4]UK Research and Innovation, Science and Technology Facilities Council, Rutherford Appleton Laboratory, Chilton, UK
[5]Institute for Energy and Climate Research: Stratosphere (IEK–7), Forschungszentrum Jülich, Jülich, Germany
[6]Laboratoire Atmosphères, Milieux, Observations Spatiales (LATMOS-IPSL), UMR CNRS 8190, Institut Pierre Simon Laplace, Sorbonne Univ./UVSQ, Guyancourt, France

**Correspondence:** Bernard Legras (bernard.legras@lmd.ipsl.fr)

**Abstract.** We use a combination of space-borne instruments to study the unprecedented stratospheric plume after the Tonga eruption of 15 January 2022. The aerosol plume was initially formed of two clouds at 30 and 28 km mostly composed of sub-micronic sulphate particles, without ashes washed-out within the first day following the eruption. The large amount of injected water vapour led to a fast conversion of $SO_2$ to sulphate aerosols and induced a descent of the plume to 24-26 km over the first three weeks by radiative cooling. Whereas $SO_2$ has returned to background levels by the end of January, volcanic sulphates and water still persisted after 6 months, mainly confined between 35°S and 20°N until June due to the zonal symmetry of the summer stratospheric circulation at 22-26 km. Sulphate particles, undergoing hygroscopic growth and coagulation, sediment and gradually separate from the moisture anomaly entrained in the ascending branch Brewer-Dobson circulation. Sulphate aerosol optical depths derived from the IASI infrared sounder show that during the first two months the aerosol plume was not simply diluted and dispersed passively but rather organized in concentrated patches. Space-borne lidar winds suggest that those structures, generated by shear-induced instabilities, are associated with vorticity anomalies that may have enhanced the duration and impact of the plume.

## 1 Introduction

The phreato-magmatic eruption of the Hunga Tonga-Hunga Ha'apai (hereafter HTHH) on 15 January 2022 was exceptional in several respects. Its explosive intensity is close to that of the eruption of Mount Pinatubo in 1991, with a Volcanic Explosivity Index of ∼6 (Poli and Shapiro, 2022). The induced atmospheric Lamb wave circled the globe at least 4 times with an amplitude comparable to that of the 1883 Krakatau eruption (Matoza et al., 2022; Vergoz et al., 2022; Wright et al., 2022). Within a few hours, several successive events injected material up to the mesosphere, with the bulk of the plume being detrained

between 26 and 34 km (Carr et al., 2022; Khaykin et al., 2022; Podglajen et al., 2022; Proud et al., 2022; Taha et al., 2022). A further remarkable fact is that the plume carried an unprecedented amount of water vapour into the stratosphere, increasing instantaneously its overall water vapour content by $\sim$10% (Millán et al., 2022; Khaykin et al., 2022). Quite surprisingly, the satellite data gathered after the event reported a stratospheric $SO_2$ injection of only 0.5 Tg, on par with much smaller and less explosive eruptions (Millán et al., 2022; Carn et al., 2022). This led to an early estimate of negligible climatic impact (Witze, 2022; Zhang et al., 2022). Here, we report on the evolution of the stratospheric plume during the first months after the eruption and we advocate that, due to the amount of water vapour and of the sulphate aerosols which have resulted from a fast conversion, its climatic effect is very significant. We focus on the circumnavigation of the plume and proceed from the large-scale to the local patterns.

## 2 Six-month evolution of the zonal mean

Figure 1 shows the zonal mean stratospheric conditions in January-March. In the domain 21-28 km and 25°S-15°N, they are characterized by an easterly band with a maximal angular speed of 30°/day (at 25 km and 5°S in Fig. 1a). The diabatic heating rate is positive everywhere except at a narrow region near 27 km over the equator (Fig. 1b). These conditions are stable during the whole January-March period (Fig. 1d-g). In April-June, the angular speed weakens and changes sign (Fig. 1d&f) while the warming turns to cooling (Fig. 1e&g) as a combined effects of the Quasi Biennial Oscillation (QBO) and the seasonal cycle.

Figure 2 shows that after an initial rapid meridional dispersion in the first days after the eruption (Khaykin et al., 2022), the aerosol and water plume stay mostly confined within the latitude band 35°S-20°N until June when wave activity increases, and evolve slowly in the zonal mean. By mid-February, aerosols and water vapour have already spread through all longitudes (Khaykin et al., 2022, , and Fig. 6 below). The OMPS-LP extinction ratio increases in time at the core of the cloud and reaches a maximum in mid-April. The simultaneous increase of the CALIOP scattering ratio suggests particle growth. The MLS water vapour distribution initially coincides with that of the aerosols but they progressively move apart vertically (see also Schoeberl et al., 2022) due to the sedimentation of the aerosols. By early June, aerosols and moisture appear fully separated, respectively below and above 25 km.

The plume vertical motion (Appendix A3) is analysed (Fig. 3) for two latitude bands and the apparent aerosol radius is estimated by interpreting the aerosol plume motion as a fall speed of the scattering particles. The descent of aerosols separates in two subsequent phases. For the first phase, lasting until about 20 February, Figure 3a-d show a fast descent in the two latitude bands which would imply unrealistically large aerosol sizes (Fig. 3e). During this phase, the water vapour follows the aerosol downward motion. Sellitto et al. (2022) (S2022 hereafter) explain this behaviour by the cooling effect of water vapour infrared emission which is strong as long as the water vapour is concentrated and is located well above its neutral radiative level. Sedimentation is then a secondary effect.

In the second phase, after 20 February, the diluted water vapour is ascending but is still producing a cooling detected by the departure of its ascent rate from the ERA5 ascent rate (Appendix A2), especially in the 25°S-15°S band, which persists until June (see also Schoeberl et al., 2022; Coy et al., 2022) where it eventually vanishes as water vapour gets too diluted to

produce significant cooling. The estimated sedimentation rate of the scattering aerosols and the corresponding particle radius (Appendix A3) suggest (Fig. 3e) that the aerosol particle size grows up to about 1.4 μm in April-May and starts shrinking in May-June in the 15°S-5°S band whereas it stays near 1 μm until June before shrinking in the 25°S-15°S band.

The extinction-to-backscatter ratio, obtained by combining OMPS-LP and CALIOP data (Fig. 3g) exhibits a growth followed by a decay which are qualitatively consistent with the aerosol size evolution (Fig. 3g) and the expected behavior of the ratio (Fig. 3f) in the 1-2, μm size range. Considering the saturation and decay of the extinction (Fig. 2), and the progressive vertical separation of aerosols and moisture, we suggest that the initial growth of the particles was by hygroscopic growth until April, where the extinction culminates, and was followed by coagulation over April-May and then by a decay due to evaporation as

the ambient air gets drier and the aerosol plume is diluted. Coagulation and evaporation are obviously not exclusive and their competition depends on the ambient conditions that vary over space and time (Hamill et al., 1977). It is also apparent from Fig. 2 that the moist layer is less confined than the aerosol layer and extends in latitude beyond the limits of the figure. The extinction-to-backscatter ratio is also smaller on the periphery of the aerosol plume (Fig. 3g). Therefore, we expect evaporation of the transported sulphate aerosols to occur at such latitudes.

## 3    Inferred composition of the plume

We now consider the history of the aerosol composition of the plume. The sequence in Fig. 4a-d shows, in agreement with Carr et al. (2022), S2022 and Khaykin et al. (2022), that the ash and ice cloud (brown and deep blue) is rapidly removed within the first day following the eruption likely via sedimentation of large ice particles which condensed water in excess of the saturated profile up to 35 km on 15 January (Khaykin et al., 2022). Taha et al. (2022) mention that ashes are missing in

the plume on 17 January from UV satellite observations. What emerges on the west side are two greenish clouds (C1 and C2 on Fig. 4b-d) without any hint of ash (ashes would appear as yellow/reddish). The CALIOP cross section through these clouds (Fig. 4e-f) shows high-scattering-ratio without depolarization, hence indicative of predominantly small spherical particles. The two clouds C1 and C2 are well separated in altitude. A few days later, the Light Optical Aerosol Counter flight and ground lidar observations, both from La Réunion, confirm this by showing sub-micron size, mainly non-absorbing, particles (Kloss

et al., 2022; Baron et al., 2022).

A further source of information is from the Infrared/Microwave Sounder (IMS) retrieval (Appendix A1.2) of $SO_2$ column and Sulphate Aerosols (SA) optical depth. Figure 4g-h shows that the conversion to sulphates started immediately after the eruption with an SA optical depth reaching 0.1 one day after the eruption suggesting that the two clouds seen by CALIOP are composed of almost pure sulphate droplets. The fast conversion of $SO_2$ to sulphate aerosols is also discussed by S2022 and

Zhu et al. (2022), using observations and chemical/transport modelling, respectively. The presence of a significant amount of gas sulphates is not expected under the ambient conditions of the plume (Hamill et al., 1977; Tsagkogeorgas et al., 2017).

Four days later (Fig. 5b), the two clouds are still separated but have elongated under the zonal shear forming a pair of long strips. Comparing Figs. 5a and b, makes apparent that the conversion to sulphates is almost complete in the western strip generated from C1 while it is incomplete in the eastern strip generated from C2. S2022 show that the western cloud C1 is much

moister than the eastern cloud C2, offering a likely reason for faster conversion, as also discussed by Zhu et al. (2022). A cloud C3 of almost pure $SO_2$ is located between Australia and Indonesia (Fig. 5a), at lower altitudes than the other two clouds, as inferred from its low traveling angular speed. Comparing the IMS products to RGB-Ash (Fig. 5c) demonstrates that RGB-Ash shows sulphates rather than $SO_2$ as usually assumed since both C1 and C2 are present but C3 is absent. The sensitivity of geostationary broad-band products, like RGB-Ash, to sulphates is shown by Sellitto and Legras (2016).

The conversion of remaining $SO_2$ to sulphates proceeds until $SO_2$ returns to background conditions by late January (Fig. 5d). The sulphates persist for at least six months (Fig. 2) and the comparison of Fig. 5e and f shows that zonal averages of IMS and CALIOP products exhibit very similar patterns. The CALIOP depolarization ratio never exceeds its initial value (Fig. 4f) until July.

## 4    Circumnavigation and instabilities

Figure 6 shows the circumnavigation of the sulphate plume from a series of SA optical depth maps over one month and half (an extended view until 30 April is provided by the supplement movie (Appendix B)). Due to the differential rotation, the fastest patches near $5°S$ caught the slowest by $30°S$ by mid-February and the plume filled the whole latitude circle. As time proceeds the components of the plume kept elongating and mixed together towards a zonal uniformity (see movie).

However, Fig. 6 shows a number of localized concentrated patches which persist and keep forming in the plume one month after the eruption. Figure 7 investigates the structure of some of them and compares the SA optical depth to the observations from active instruments. Using the Atmospheric Laser Doppler Instrument (ALADIN) (Appendix A1.5), we see (Fig.,7b), on 24 January, an anomalous anticyclonic shear across the highest (28 km) patch at $5°W$ and $22°S$ in Fig.7a which is part of the western strip defined in Sec. 3. The same pattern is observed on 28 January (Fig. 7c) across a patch near $11°E$ and $25°S$ (Fig. 6a) which belongs to the eastern strip. The corresponding CALIOP section (Fig. 7d) exhibits a "jelly fish" pattern with a head at $26\,km$ connected by a tail to lower altitude patches along an arc of same angular speed (Fig. 1a). This pattern is found repetitively on subsequent CALIOP sections (not shown).

On 30 January 2022, we are back on the western strip (Fig. 7e) and the corresponding CALIOP section (Fig. 7f) shows filaments overlying the main compact patch that we interpret as a tail left by the fast descent. Again this pattern is repetitively observed on CALIOP sections across the western strip until mid-February where the fast descent halts. The western strip originates from the moistier cloud in Fig. 5e that descended faster than the other.

A remarkable feature in the SA optical depth maps is the train of compact elliptical structures linked together by filaments which is visible all along February and early March in Fig. 6 and the supplemental movie. This peculiar shape is reminiscent of shear-induced instabilities (Juckes, 1995), leading to the formation of a chain of vortices. The suspicion is reinforced by the pattern of a wrapping-up tripolar structure seen near $180°E$ on 11 February (Fig. 7g) and perfectly captured by CALIOP as a core surrounded by two arms at the same level (Fig. 7h). This comparison also reveals the ability of the IMS product to retrieve small-scale details.

Barotropic shear instability requires a reversal of the meridional gradient of absolute vorticity. The mean flow in ERA5 does not satisfy this criterion. A generalized baroclinic instability requires a reversal of the potential vorticity gradient but the mean flow again hardly satisfies this criterion at the required altitude of 25 km (Fig. 8). The very fact that the instability produces aerosol patches suggests that they are related to the generation of vorticity. The detection of an anomalous anticyclonic shear across the concentrated patches of the plume by ALADIN (Fig. 7b-c) supports this hypothesis. However, sulphates are poor absorbers and neither these vortical structures nor their thermal signature have been detected by our present investigation of the ERA5. Therefore, this observation still requires an explanation that we leave for future studies.

## 5    Discussion and conclusion

The very intense and unusual HTHH eruption generated an intense and unusual stratospheric plume with a huge amount of injected water vapour that remains well above normal 6 months after the eruption. After a fast initial removal of ice and ashes, the bulk of the remaining plume consisted of two main clouds between 26 and 32 km traveling westward due to the prevailing phase of the QBO. The ensuing zonal transport dispersed the plume through all longitudes in less than a month (see also Khaykin et al., 2022). The initial $SO_2$ is fully converted into sulphates in less than two weeks under the influence of water vapour. Notice that the absence of detection does not entirely rule out the possible presence of very thin ash as nucleous within sulphate liquid droplet without optical signature.

The fast initial descent of the upper part of the plume induced by the radiative water vapour cooling has concentrated the aerosols within a fairly narrow layer, about 2 km thick as seen from CALIOP measurements (Figs. 2&5). Within the limit of MLS resolution, the water vapour distribution then coincides with the aerosols. The aerosols later continued subsiding at a slower rate under the effect of gravitational sedimentation, whereas the moist layer entrained by the Brewer-Dobson circulation was simultaneously ascending, so that the two layers progressively separated (as also seen by Schoeberl et al., 2022). The spurious warming in the ERA5 that overlaps the moist layer suggests that radiative cooling by water vapour persists until May (see also Schoeberl et al., 2022; Coy et al., 2022). Although a precise sequencing is difficult without quantitative modelling, it is likely that the sulphate aerosols first grew by hygroscopic growth, then by coagulation and ended by dwindling under evaporation. Our estimation of fall speed and extinction-to-backscatter ratio trends is consistent with a growth up to about 1.4 μm and then a decrease in mean size.

The fast conversion of $SO_2$ suggests that the initial sulphur injection might have been underestimated. Consistently, S2022 showed that the HTHH eruption produced the largest stratospheric aerosol perturbation since the Pinatubo eruption in 1991, and suggested a large potential for climatic impacts (see also Khaykin et al., 2022). By June, the hemispheric stratospheric aerosol optical depth perturbation of the HTHH plume is twice as large as the peak perturbation of the 2019 Raikoke eruption, and the tropical impact is at least three times as large as any volcanic perturbation since Pinatubo 1991 (S2022, Khaykin et al., 2022). As the $SO_2$ emissions for the Raikoke eruption have been estimated at 1.5 Tg (de Leeuw et al., 2021), we assume this value as the lower limit for the HTHH eruption, three times larger than early estimates (Witze, 2022). The young aerosols seem mostly made of sub-micronic liquid sulphate particles then growing to 1.4 μm due to hygroscopic growth and coagulation. The

dispersion of the plume questions the magnitude and the duration of the impact. An early estimate of the resulting radiative

forcing by S2022 shows that stratospheric aerosol and water vapour perturbations from the eruption may significantly impact the climate system. Given the large greenhouse potential of stratospheric water vapour (e.g. Solomon et al., 2010), it was proposed that the dispersed plume has a net warming effect (S2022), in contrast with the cooling expected from stratospheric aerosols.

Finally, we have shown that the dynamics repetitively generates compact aerosol structures in a process that bears similarities with shear instability and that some structures carry anomalous anticyclonic vorticity. That points to the possible role of such processes in extending the life time and the impact of the plume.

**Appendix A: Data and methods**

**A1 Observations**

We use data from the following instruments and products.

**A1.1 CALIOP**

The Cloud-Aerosol Lidar with Orthogonal Polarisation (CALIOP) is a spaceborne lidar onboard the CALIPSO satellite (Vaughan et al., 2004; Winker et al., 2010). We use the L1 $532\,nm$ attenuated backscatter which is filtered in the horizontal with a median filter of width $102\,km$. In particular, this filter removes the noise associated with the South Atlantic Anomaly

(SAA) in the Earth's magnetic field which perturbs CALIOP data between $30°W$ and $80°W$ (Noel et al., 2014). In practice, a limited amount of data are usable in this region and only at night. After filtering, the data are further averaged at a resolution of $34\,km$ for compactness. The other channels are processed in the same way.

Due to solar activity, CALIOP was not operating on 18 January and between 20 and 26 January. Hence, our CALIOP series start on 27 January. We use only night data in this work. The molecular backscatter is calculated following Hostetler et al.

(2006). For each day, the backscatter ratio is zonally averaged over all available orbits of that day (14 to 15 for a nominal day). The native vertical resolution in the 20-30 $km$ range is $180\,m$.

**A1.2 IMS**

The RAL (Rutherford Appleton Laboratory) Infrared/Microwave Sounder (IMS) retrieval core scheme (Siddans, 2019) uses an optimal estimation spectral fitting procedure to retrieve atmospheric and surface parameters jointly from co-located measure-

175 ments by IASI (Infrared Atmospheric Sounding Interferometer), AMSU (Advanced Microwave Sounding Unit) and MHS (Microwave Humidity Sounder) on MetOp-B spacecraft, using RTTOV 12 (Radiative Transfer for TOVS)(Saunders et al., 2017) as the forward radiative transfer model. The use of RTTOV 12 enables the quantitative retrieval of volcanic-specific aerosols (sulphate aerosol) and trace gases ($SO_2$). The present paper uses IMS $SO_2$ and sulphate aerosols observations from its near-real time implementation. The IMS scheme retrieves the $SO_2$ in the sensitive region around $1100$-$1200\,cm^{-1}$, in ppbv assuming

a uniform vertical mixing ratio. It retrieves sulphate-specific optical depth at $1170\,\mathrm{cm}^{-1}$ (i.e. the peak of the mid-infrared extinction cross section (Sellitto and Legras, 2016)), assuming a Gaussian extinction coefficient profile shape peaking at $20\,\mathrm{km}$ altitude, with $2\,\mathrm{km}$ full-width half-maximum. The bulk of the spectroscopic information on $SO_2$ and sulphate aerosols, in the IMS scheme, thus comes from the Infrared Atmospheric Sounding Interferometer (IASI) (Clerbaux et al., 2009). We refer to the two retrieved product as IMS $SO_2$ and SA optical depth (SA OD in the figure titles) in this work. The data are provided

daily on a regular grid with $0.25°$resolution in latitude and longitude with one image collecting the day swaths and another collecting the night swaths.

### A1.3 OMPS-LP

The Ozone Mapping and Profiler Suite Limb Profiler (OMPS-LP) onboard the Suomi-NPP satellite provides along track vertical profiles of aerosol extinction in several visible bands (Loughman et al., 2018; Taha and Loughman, 2020). We use version

2.1 and the 745 nm band as recommended by Taha et al. (2021). Swaths with non zero quality flag are discarded. Basically, this filters data polluted by the SAA but filtered and non filtered results differ very little in our processing. The molecular extinction is calculated from the same formulas as the CALIOP molecular backscatter but for a change of wavelength. The extinction is averaged daily over all available orbits of that day and after a horizontal interpolation to a latitude grid of $1.1°$resolution that corresponds to the mean resolution of OMPS-LP in the considered range of latitudes.

OMPS-LP has a vertical resolution of $1.5\,\mathrm{km}$ which is lower than the vertical resolution of CALIOP. It is also sensitive to the arch effect (Gorkavyi et al., 2021) for that tends to shift downward by several kilometers the apparent bottom of an extended aerosol layer.

### A1.4 MLS

The Microwave Limb Sounder (MLS) onboard NASA's AURA satellite provides along track vertical profiles of water vapour

mixing ratio (Lambert et al., 2015; Schwartz et al., 2020) as well as other trace gases, temperature and cloud ice. We use the version 4 without accounting the quality flag as in Millán et al. (2022). The data are projected and zonally averaged daily onto a fixed latitude grid of $1.45°$resolution in the domain of interest. As they are provided on pressure levels with an approximate vertical resolution of $1.5\,\mathrm{km}$, similar to that of OMPS-LP, they are interpolated to altitudes using the geopotential calculated daily on the ERA5 zonal mean. In order to get estimates of the altitudes by the method described in Appendix A3,

the interpolation is made to a resolution of $100\,\mathrm{m}$ using a non-oscillating Akima interpolator. The native vertical resolution of the product is about $1.5\,\mathrm{km}$ in the relevant range of altitudes.

### A1.5 ALADIN

The Atmospheric Laser Doppler Instrument (ALADIN) onboard the Aeolus satellite is the first space-borne Doppler wind lidar. It is designed to measure wind along the line of sight from the Doppler shift of the 355 nm light emitted by the laser and

210 scattered back by molecules (Rayleigh wind) or aerosols (Mie wind). Horizontal line-of-sight wind is retrieved neglecting the

vertical wind component. The anomaly wind is calculated by removing the background wind at same time and location from ERA5. As the line-of-sight is perpendicular to the heliosynchronous orbit, the measured component at low and mid-latitudes is essentially the zonal wind. The ceiling of Aeolus vertical bins can be adjusted and was increased to $30\,\mathrm{km}$ in the area of the HTHH plume ($30°\mathrm{S}$-$0°$) a few days after the eruption. We use the Mie product which is of better quality than the Rayleigh product inside the plume Zuo et al. (2022).

### A1.6 RGB-Ash

We use a composite RGB product, denoted as RGB-Ash, that benefits from the sensitivity of the $8.5\,\mu\mathrm{m}$ band of the Advanced Himawari Imager (AHI) and Spanning Enhanced Visible and InfraRed Imager (SEVIRI) onboard the geostationary Himawari-8 and Meteosat-8 satellites. The product is based on the EUMETSAT Ash RGB recipe (https://navigator.eumetsat.int/product/EO:EUM:DAT:MSG:VOLCANO/print) and uses the brightness temperatures (BT in K) of the three channels: 8.5, 10.4 and 12.3 μm. The recipe for the three colour indexes ranging from 0 to 1 is $R = (\mathrm{BT}(12.3) - \mathrm{BT}(10.4) + 2574)/6$, $G = (\mathrm{BT}(10.4) - \mathrm{BT}(8.5) + 4)/9$ and $B = (\mathrm{BT}(10.4) - 243)/60$. The same recipe is used for both instruments even if the channels are not strictly identical. This product allows to qualitatively distinguish thick ash plumes or ice clouds (brown), thin ice clouds (dark blue) and sulphur-containing plumes (green). Mixed ash/sulphur-containing volcanic species would appear in reddish and yellow shades.

### A2 ERA5 reanalysis and meteorological data

We use the European Center for Medium Range Forecasts ERA5 reanalysis (Hersbach et al., 2020) at $1° \times 1°$ resolution and all the model levels with 6-hourly sampling at 0, 6, 12 and 18 UTC. Geopotential, potential temperature and potential vorticity are calculated at full resolution for each time. All the fields are then averaged in longitude and over the four daily samples to provide a daily zonal average. At the stratospheric altitudes which are relevant to this study, the model levels are pure pressure and therefore the averages are made over isobars.

The total all sky radiative heating rate is converted into diabatic vertical velocity from the profile of pressure, geopotential and temperature. The motion of the isentrops with respect to the geopotential in the zonal mean is used to define the adiabatic vertical velocity (see Appendix A3).

The ERA5 does not assimilate the anomalous water vapour or the aerosols in the stratosphere and therefore cannot account for their direct radiative effect, either shortwave absorption or longwave absorption and emission. However, it assimilates the induced temperature perturbation if large enough to be detected and then react to damp it by longwave radiative relaxation with a time-scale of the order of 5 days at 25km (Lestrelin et al., 2021).

In the present case, the water vapour radiative cooling creates a negative temperature anomaly overlapping the plume Schoeberl et al. (2022); Coy et al. (2022) that genearates a spurious compensating heating rate and exaggerated vertical ascent.

The Lait potential vorticity (LPV) used in Fig.8 is defined from the Ertel potential vorticity (PV) as

$$\mathrm{LPV} = \left(\frac{600}{\theta}\right)^4 \mathrm{PV},$$

where $\theta$ is the potential temperature in $K$.

## A3 Vertical motion from CALIOP and MLS

The observed vertical motion is obtained from CALIOP and MLS by applying a second-order Savitsky-Golay filter with a 31-day window to the daily mean vertical location of CALIOP scattering ratio and MLS water vapour, retaining data above 2 and 6 ppmv offsets, respectively. The offset are defined to isolate the aerosol and the water plumes from the background. The 31-day window has been adjusted from several trials with 11, 21, 31 and 41 days as the value beyond which the resulting motion curve was rid of short time fluctuations and did not change any more in shape. This was reached with the 31-day window for MLS and the 21-day window for CALIOP but in the sake of consistency we use the 31-day window for both.

The diabatic and adiabatic background vertical velocities are calculated from ERA5 zonal means. The diabatic motion results from the total radiative heating rate $\frac{DT}{Dt}|_{\text{RAD}}$ multiplied by $\frac{\theta}{T}\frac{\delta z}{\delta \theta}$ where $(T, \theta, z)$ are temperature, potential temperature and geopotential altitude. The adiabatic motion, which is always a small correction, is estimated as $-\frac{\partial \theta}{\partial t}|_p \frac{\delta z}{\delta \theta} + \frac{\partial z}{\partial t}|_p$. The calculations are made by centered finite differences on the model grid which is in pure pressure in the considered altitude range.

Two correction methods are applied to the observed descent of aerosols from CALIOP in order to estimate the sedimentation velocity with respect to the air. In the first method, the ascent of the water vapour plume seen by MLS is used as an estimate of air motion and in the second method, the sum of diabatic and adiabatic ERA5 vertical velocities provides this estimate. Then this air motion is added to the aerosol descent rate to estimate the sedimentation velocity. The first method is applicable to the period during which the aerosol and water vapour distributions overlap and the water vapour cooling perturbs the heating rate estimate of ERA5. The second method applies at a later stage when the aerosol and water vapour distributions are well separated and the radiative effect of water vapour has been defeated by dilution. As the boundary between these two regimes cannot be defined accurately we show the results of the two methods.

The scattering aerosol radius is then estimated using the Stokes fall speed formula for small particles (Seinfeld and Pandis, 2016).

## A4 Mie calculations

The theoretical extinction-to-backscatter ratio for the plume has been calculated using the Python-based miepython Mie code, available at: https://miepython.readthedocs.io/en/latest/. The extinction and backscatter coefficients have been estimated at 750 and 532 nm, respectively, to simulate OMPS and CALIOP observations. Typical sulphate aerosols refractive indices have been considered, with the assumption of very weakly absorbing particles (based on the results of (Kloss et al., 2022)). Log-normal size distributions with varying standard deviation are simulated, to study how this ratio changes with radius.

## Appendix B: IMS animation

The supplement animation https://doi.org/10.5281/zenodo.7102472 shows the IMS SA optical depth product for all day and night orbits of each day between 13 January and 30 April 2022. The indicated times are those of the intersection of the orbits with the equator. When two orbit swaths overlap, the crossing time of the overlapped orbit is indicated in red. Missing orbits are blanked out. Several days are entirely missing between 8 and 14 March.

*Code and data availability.* MLS and OMPS-LP data are available from EarthData centre at: https://disc.gsfc.nasa.gov/. CALIOP data v3.41 are available at: https://doi.org/10.5067/CALIOP/CALIPSO/CAL_LID_L1-VALSTAGE1-V3-41. ALADIN data are available from ESA at https://earth.esa.int/eogateway/missions/aeolus/data. IMS data are available at: https://doi.org/10.5281/zenodo.7102472. ERA5 data are available at https://www.ecmwf.int/en/forecasts/datasets/reanalysis-datasets/era5. The python scripts and notebooks used in this study are available at https://github.com/bernard-legras/ASTuS.

*Video supplement.* https://doi.org/10.5281/zenodo.7102472

*Author contributions.* BL, PS and AP conceived the study and conducted the analyses. EC and RS elaborated and provided the IMS product. JUG, SK and FP were involved in discussions of the results and their interpretation. The paper was written by BL, CD, PS and AP. All authors contributed to the final version.

*Competing interests.* The authors declare no competing interests.

*Acknowledgements.* The CALIOP and GEOs data have been provided by the AERIS/ICARE Data and Service Centre and processed using the centre computer resources. ERA5 data were provided by Copernicus Climate Change Service. We acknowledge support from the Agence Nationale de la Recherche under grants 21-CE01-0007-01 (ASTuS), ANR-21-CE01-0016-01 (TuRTLES) and 21-CE01-0028-01 (PyroStrat). We acknowledge discussions with Paul Billant, Guillaume Carazzo, Corinna Kloss, Guillaume Lapeyre, Angela Limare and Tjarda Roberts.

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

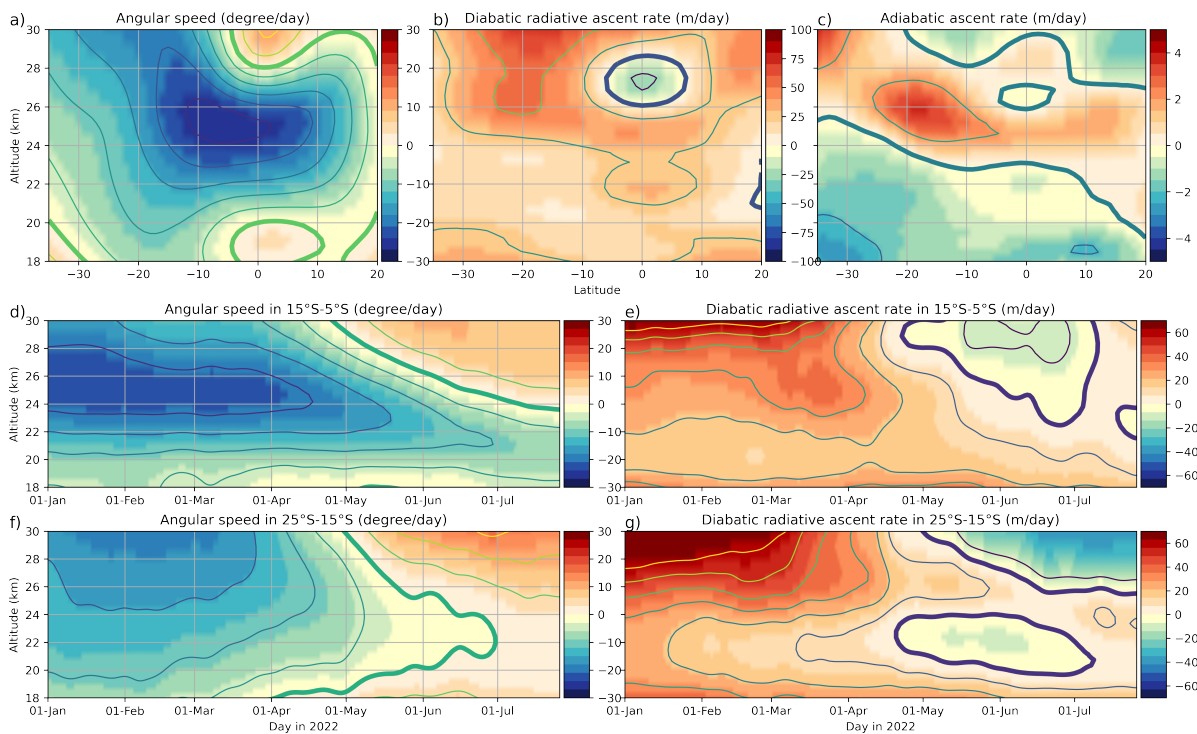

**Figure 1.** (a): Zonal mean angular rotation speed $\omega = \frac{U}{R_{\text{Earth}}\cos(\phi)}$ from ERA5, averaged between 15 January 2022 and 15 March 2022 (in degree day$^{-1}$). (b): Same as panel (a) for the diabatic ascent rate calculated from the total all-sky ERA5 heating rate (in m day$^{-1}$). (c): Same as panel (a) for the adiabatic ascent rate due to motion of the isentropic surfaces with respect to the geopotential surfaces (in m day$^{-1}$). (d): Daily zonal and altitude band average angular speed between $15°$S and $5°$S as a function of time (in degree day$^{-1}$). (e): Same as panel (d) for the diabatic ascent (in m day$^{-1}$). (f-g): same as panels (d-e) for the latitude band between $25°$S and $15°$S.

OMPS-LP 745 nm daily zonal average aerosol extinction ratio
CALIOP daily zonal average 532 nm attenuated aerosol scattering ratio
MLS water vapour daily zonal average mixing ratio (ppmv)

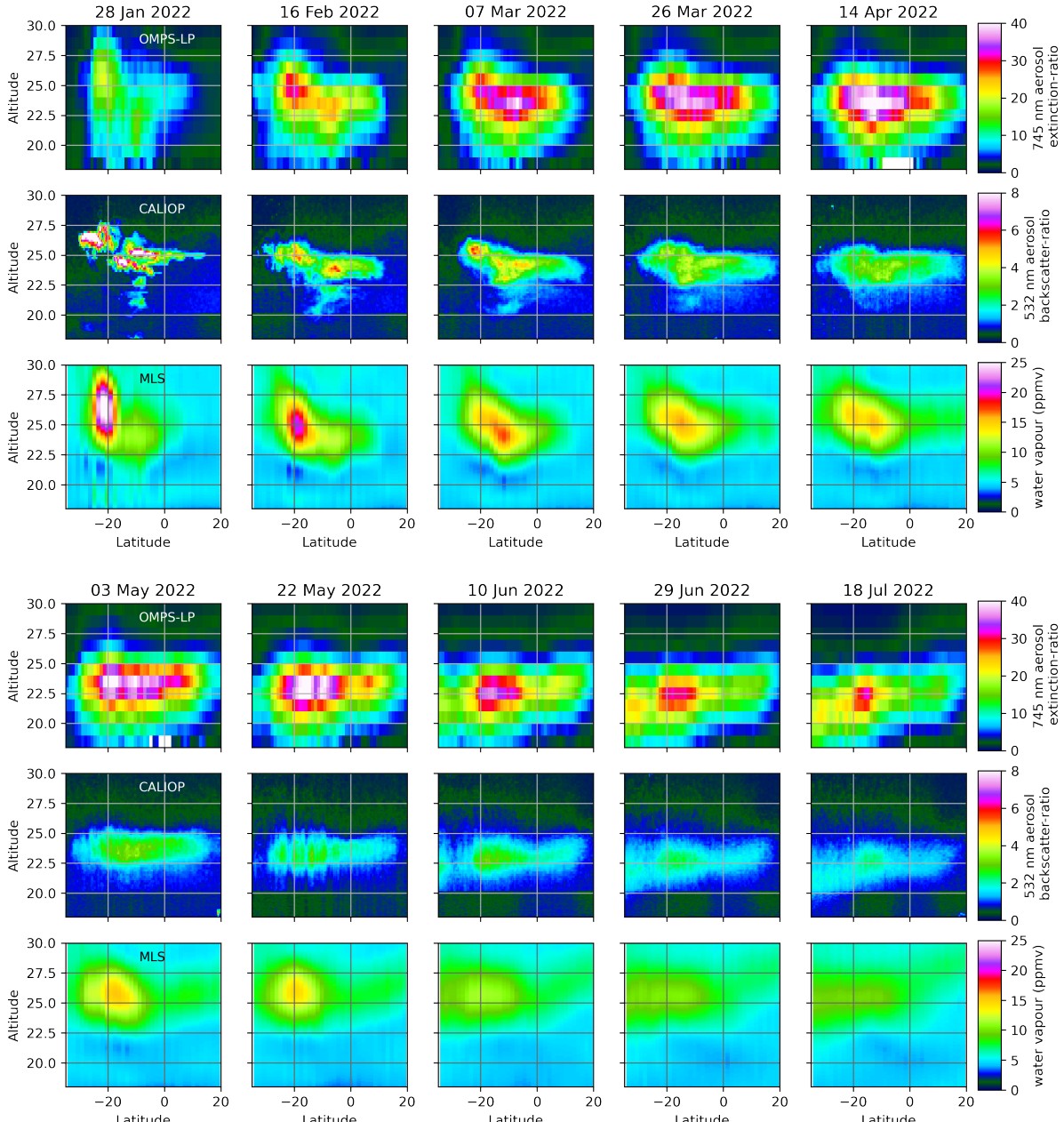

**Figure 2.** Series of daily zonal averages over all available orbits from three satellites measuring aerosols and water vapour (Appendices A1.3, A1.1 and A1.4). The series is shown in two consecutive blocks of three rows. (Upper row): OMPS-LP 745 nm aerosol extinction ratio. (Middle row): CALIOP 532 nm aerosol attenuated backscatter ratio. (Lower row): MLS water vapour (in ppmv). Days from 28 January 2022 to 16 July 2022 with 19-day step.

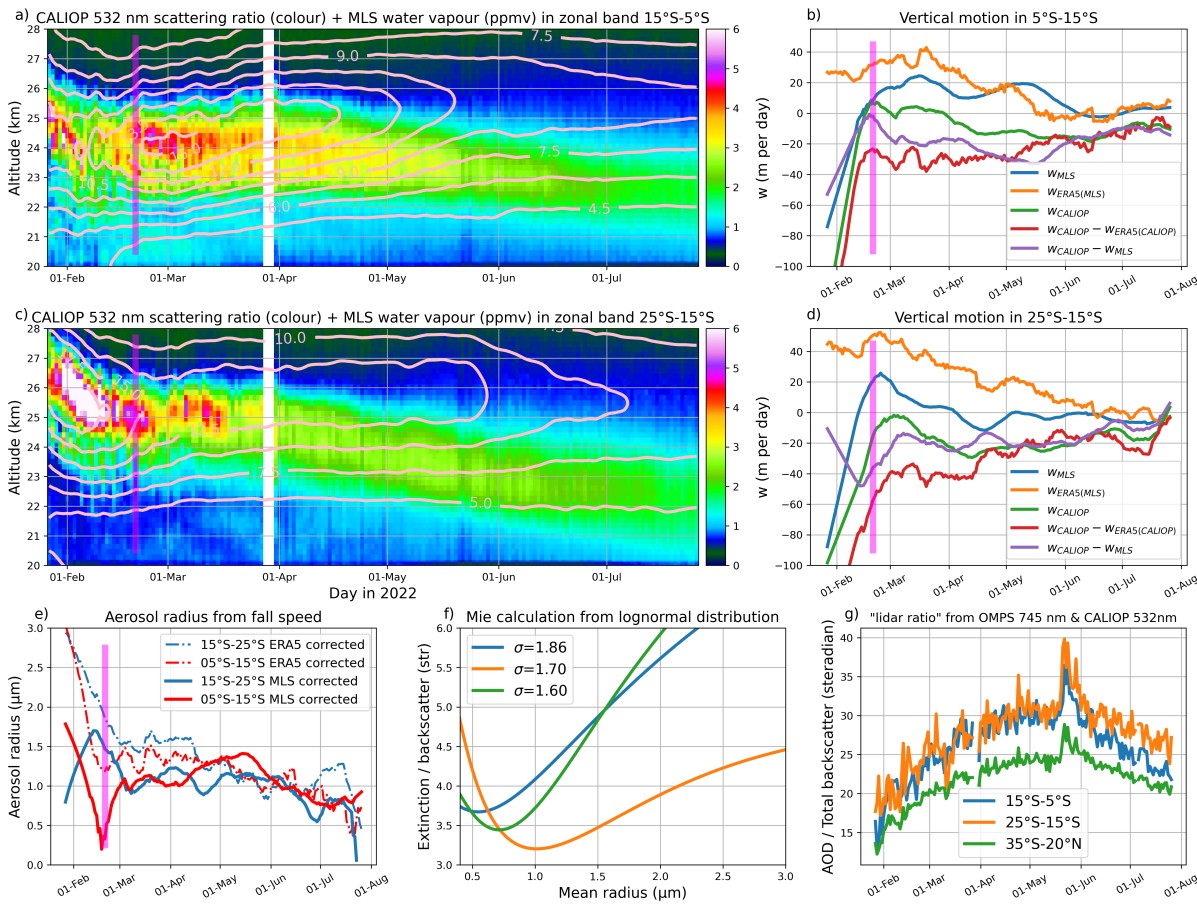

**Figure 3.** (a) and (c): Zonal and latitude band averages as a function of time for CALIOP 532 nm scattering ratio (colour) and MLS water vapour (contours, ppmv) for the 15°S-5°S and 25°S-15°S latitude bands, respectively. (b) and (d): Vertical motions for the same two latitude bands: $W_{CALIOP}$ for aerosol plume and $W_{MLS}$ for water vapour plume deduced from (a) and (c) (Appendix A3), $W_{ERA5(MLS)}$ for ERA5 ascent rate (Appendix A2) at the location of the MLS plume, $W_{CALIOP} - W_{ERA5(CALIOP)}$ and $W_{CALIOP} - W_{MLS}$ for the CALIOP sedimentation speed estimated, respectively, with respect to the ERA5 ascent rate and the MLS water ascent rate (Appendix A3). (e): Aerosol radius deduced from the aerosol sedimentation speed interpreted as an aerosol fall speed and using Eq. (9.42) of Seinfeld and Pandis (2016). The dash lines show the estimate for the ERA5 correction which is applicable after May when the aerosol and moist layers are separated and the water vapour cooling has ceased and the solid lines show the estimate for the MLS correction which is valid as long as the aerosol and moist layers overlap (Appendix A3). (f): Ratio of the theoretical 745 nm aerosol extinction and 532 nm aerosol backscatter cross sections, calculated using a Mie code (see Appendix A4) with three values of the standard deviation $\sigma$. (g): Ratio of the 745 nm OMPS LP aerosol optical depth and 532 nm CALIOP integrated attenuated backscatter, both over the vertical range 18 to 30 km. The curves are shown for the same latitude bands as in a) and c) and for the 35°S-20°N band that encompasses also the periphery of the aerosol plume. In panels (a-e), a vertical line is drawn on 20 February to indicate the separation between the two phases of the vertical motion as discussed in Sec. 2. During the last third of May, both OMPS LP and CALIOP are perturbed by the intensified solar activity and the peak seen in panel (g) at such dates must be considered as spurious.

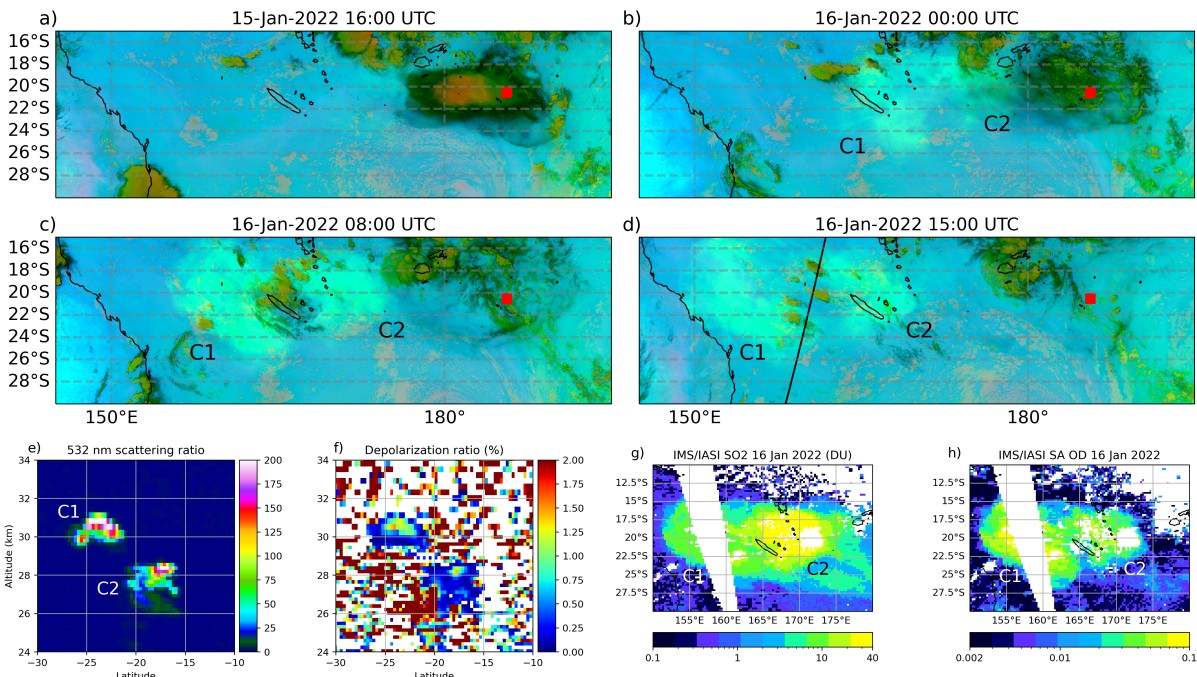

**Figure 4.** (a-d): RGB-Ash composite (see Appendix A1.6) from Himawari-8 at four selected times during the first day and half following the eruption. The red square denotes the location of the volcano. This product allows to qualitatively distinguish thick ash plumes or ice clouds (brown), thin ice clouds (dark blue) and sulphur-containing plumes (green). Mixed ash/sulphur-containing volcanic species would appear in reddish and yellow shades. (e): CALIOP 532 nm backscatter ratio at 15:08 UTC along the orbit track shown in panel (d). (f): 532 nm depolarization ratio (orthogonal channel / total) for the same orbit. (g-h): $SO_2$ and SA optical depth from IMS on 16 January 2022 for two night orbits crossing the equator at 10:26 UTC (right swath) and 12:08 UTC (left swath). The two sulphur clouds produced by the eruption are marked as C1 and C2 in all panels where they are visible but (f).

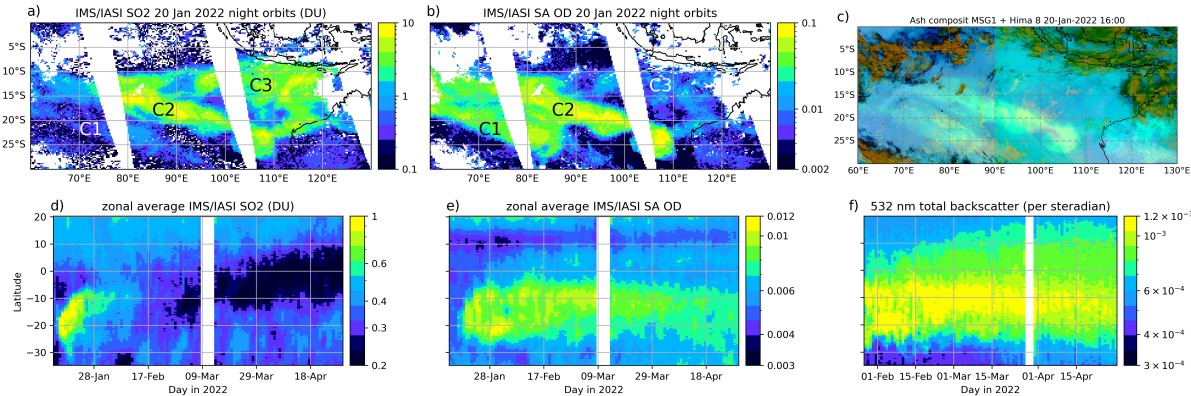

**Figure 5.** (a-b): SA optical depth and $SO_2$ from IMS on 20 January 2022 for three night orbits crossing equator at 14:06 UTC (right swath), 15:48 UTC (middle swath) and 17:29 UTC (left swath). (c): RGB-Ash composite from Meteosat-8 and Himawari-8 at 16:00 UTC on the same day. (d-e): Zonal average SA optical depth and $SO_2$ from 13 January 2022 to 30 April 2022. (f): CALIOP 532 nm attenuated backscatter integrated between 18 and 30 km from 27 January 2022 to 30 April 2022 (per steradian). The sulphur clouds C1 and C2 in Fig. 4 are now seen as two elongated strips which are marked in panels (a-b). A third cloud C3 is marked in panels (a-b).

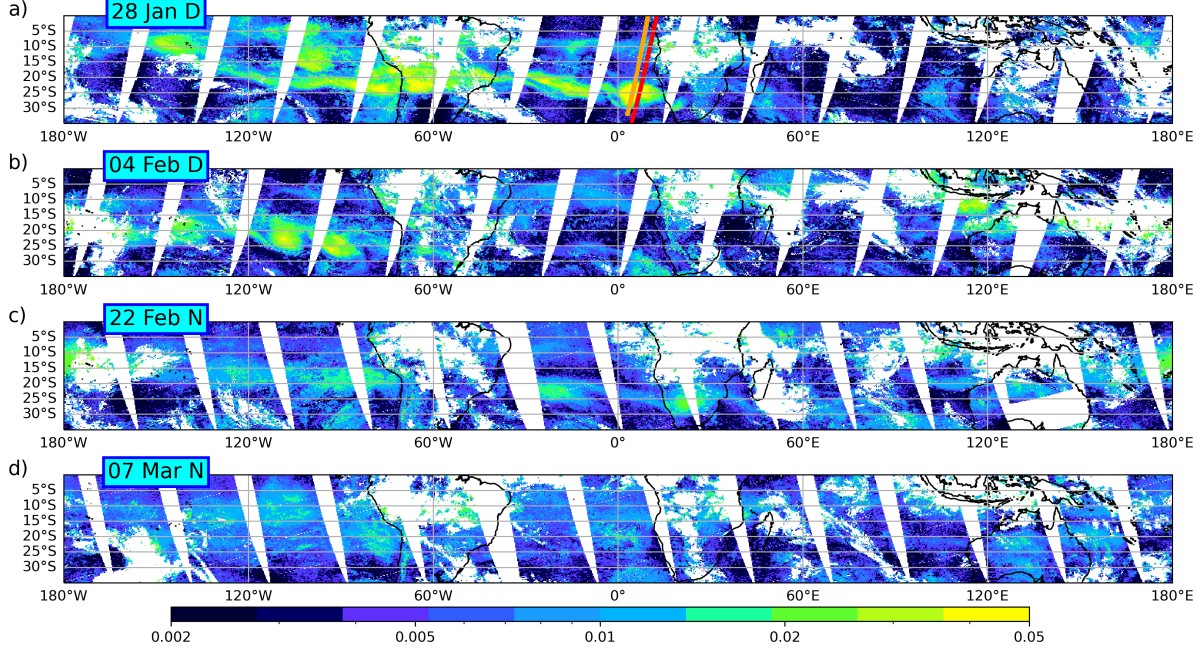

**Figure 6.** SA optical depth from IMS in the latitude range $0°$-$35°$S at four different dates as indicated. Panels (a-b) are drawn for day swaths whereas panels (c-d) are drawn for night swaths. The time progresses from right to left and the interval between two adjacent swaths is about 1h52min.

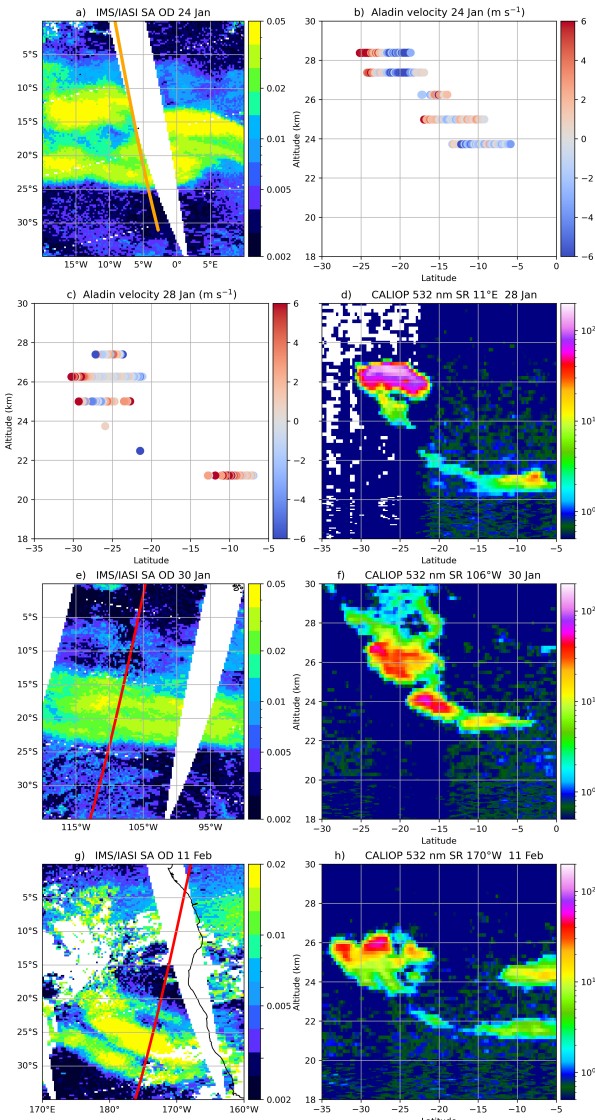

**Figure 7.** (a): IMS SA optical depth chart on 24 January 2022 near 22:52 UTC for the left swath. (b): ALADIN wind anomaly on 24 January 2022 near 18:36 UTC along the track shown in panel (a). (c): ALADIN wind anomaly near 5:12 UTC on 28 January 2022 along the the orange track shown on Fig. 6a within the IASI 8:48 UTC swath on the same day. (d): CALIOP 532 nm scattering ratio on 28 January 2022 near 1:48 UTC along the red track on Fig. 6a. (e): IMS SA optical depth chart on 30 January 2022 near 11:28 UTC for the left swath. (f): CALIOP 522 nm scattering ratio on 30 January 2022 near 9:37 UTC along the red track on panel (e). (g): IMS SA optical depth chart on 11 February near 9:49 UTC for the left s. (h): CALIOP 522 nm scattering ratio on 11 February near 13:52 UTC along the red track on panel (g).

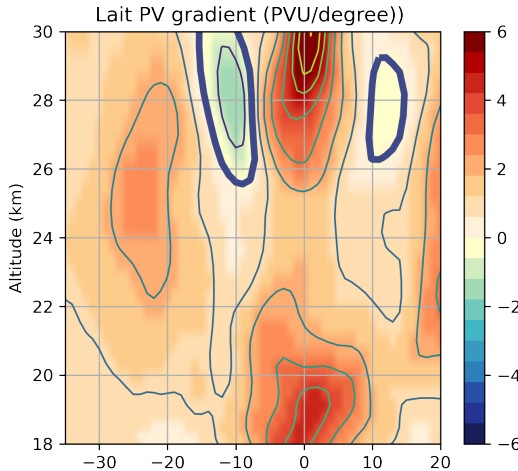

**Figure 8.** Meridional gradient of the zonal and time averaged Lait PV defined in Appendix A2. The unit is PVU per degree where 1 PVU = $10^6\,\mathrm{m^2Ks^{-1}kg^{-1}}$.