# Peer review of "The evolution and dynamics of the Hunga Tonga-Hunga Ha'apai sulphate aerosol plume in the stratosphere"

_EGUsphere, 2022_

## Author Response (AR1)

**Answer to referee 1**:

We thank the referee for his/her thorough review of our work and the numerous comments. There was initially a misunderstanding and our work has not been evaluated as a letter but as a standard submission to ACP. Therefore there are a number of requests made by the referee that could not be answered without infringing the constrains or getting away from the spirit of a letter format. However, we have been able to make a number of small modifications that meet most of the requests of the referee.

*Major comments*

*The flow of the science discussion needs to be changed. Currently the authors discuss how the plume evolve in time from January to June in section 2. Then, they discuss the composition of the plume in section 3 but this section mostly uses data from the first days after the eruption. Lastly, they discuss the circumnavigation of the plume over one month and half.*

*I suggest the authors discusses (current) section 3 first, the (current) section 4 next, and the (current) section 2 last. I also suggest that the authors add in the introduction a small plan for the study, for example: In this study we first discuss the composition of the aerosol plume focusing on the days following the eruption, then we discuss how the aerosol plume circumnavigate the globe in the following days (months), and lastly, we discuss the zonal mean patterns.*

Although another choice of presentation is possible, e.g. by discussing data as they appear in chronological order, we have chosen an order from general to more specific and local issues, which is to our opinion equally valid and fitting well within a letter format where the global pattern is presented before entering into the details. This choice allows to show in the first figure our major result on the separation between aerosols and water vapour, and it is also in the spirit of a letter not to delay the key results.

*Through the study the authors refer to the plume meaning the sulfate aerosol plume but they also discuss the water vapor plume as well as the SO2 plume. They need to be explicit through-out the manuscript about which plume are they discussing, for example the title should probably be changed to: The evolution and dynamics of the Hunga Tonga sulfate aerosol plume in the stratosphere*

We have tried to clarify when the confusion is possible. We are not restrained to the study of aerosol sulphates as recognized by the referee. We use water data and the quantitative discussion of the differences between the behaviours of the aerosol and water vapour plume is a main point of the paper. We also show both SO2 and sulphate charts. Therefore, we do think that the title deserves to be modified in the way suggested by the referee. However, the title has been modified according to the suggestion of the second referee.

*The entire section A3 requires more explanation and more details, why 21-day window and not 11 or 31 etc, is there a difference? How was this value selected? The same goes for the 2 and 6 ppmv offsets.*

*Has this been done before? If so, how did it? Are they any citations to be used. Are the ERA5 fields interpolated to the measurements time and locations, are the authors using 12UT OUT, an average of all synoptic times, etc.*

The reply regarding the window width can be found below within the reply to the comment on the two bumps in the MLS water motion (comment to p3L60). The offsets are chosen to isolate the plume from the background. This has been added to appendix A3.

This method is a natural and straightforward approach which is similar, e.g., to the numerous studies of the stratospheric upwelling due to the Brewer Dobson circulation based on satellite observations. Figure 2a&c are, technically, nothing else than Hövmoller plots. A variety of filtering methods can be used and they are all good as long as the result is robust. We use daily means of ERA5 data calculated from the four synoptic times at 0, 6, 12 and 18 UT and zonally averaged.

*P5L130-L135. All this is really interesting but the authors have not anything to support this. Is this shown in S2022? If it is please cite S2002 again so it is clearer. If this is not, please provided evidence to support these statements or delete this discussion.*

These results are indeed established in S2022 and in Khaykin et al., 2022, both works being complementary to our own with a number of common authors.

*All the data descriptions require more details, for example resolutions, footprints, validations, validity on the hunga tonga plume conditions, error estimates, etc. Please be consistent in the amount of details through all datasets.*

We use only validated standard products and, in the specific case of this eruption, we follow the recommendation of the main investigators for each instrument when they are available. For instance Millan et al. (2022) recommend to use the version 4 data instead of the version 5 and to bypass the quality flag. Taha et al. (2022) states that the version 2.1 of the OMPS-LP extinction product can be used up to 36 km. We use L1 data from CALIOP in the version 3.4 since version 4 is only available with a delay of about 7 months. There is no reason to believe that any signal above 20 km is due to anything else than aerosols and that the lidar measurements are not performed at their nominal accuracy. We use the operational product of AEOLUS that is assimilated by ECMWF.

We have improved the description of  data processing, in particular about the averaging procedures.

Minor comments:

*P1L1 change stratospheric plume to stratospheric  sulfate aerosol  plume*

We consider also the water vapour content and SO2 in the initial stage and therefore we are not restricted to sulphates although this is clearly the main topic of this work. The word limitation of the letter format does not allow to be fully descriptive in each statement, especially in the abstract.

*P1L5 the phrase "The sulphate plume persisted until June" could be interpreted as if the plume flush out of the stratosphere in June please change to " the sulphate plume is mainly confined between 35S and 20N in June (due to the zonal symmetry of the summer stratospheric circulation at 24-25 km)".  Or something along those lines.*

The manuscript has been submitted on 20 June 2022 and this was the state of the available observations. The time axis has now been extended to 26 July as requested by the second referee. The statement is still basically true as the winter wave activity in the southern stratosphere has generated meridional dispersion and mixing from  the middle of June. It has been adapted to mention the plume persistence over 6 months. Our charts are still bounded at 35S because we do not have room to extend the discussion towards the influence on mid and polar latitudes in this work.

*P1L6. Swap the order of the latitudes, that is, "between 35S and 20N".   (latitude ranges are typically expressed from the latitude further to the south to the latitude further to the north)*

Done as suggested.

*P1L17 Podglajen 2022 is not a good reference for this, use Carr et al 2022 and Proud et al 2022*

Podglajen et al. show how acoustic wave emission corresponds to plume top height, demonstrating a complex injection sequence which is not mentioned in Carr et al. 2022. It is mentioned but not characterized in Proud et al. 2022 and discussed in Khaykin et al 2022. A further study by Taha et al. 2022 shows plume height based on OMPS-LP. We are now quoting all these works together.

*P1L20. The SO2 injection of only 0.5Tg needs a citation, the authors could use: Millan et al 2022 and Carn et al 2022*

Done as suggested although Millan et al. 2022 only says that the amount of SO2 is unexceptional and does not provide a quantitative estimate.

*P2L21 the authors could add the reference Zhang et al 2022 to the Witze 2022*

Done as suggested

*P2L26 Move Figure A1 to the main body*

Because of the letter format and because this figure contains information we consider as ancillary with respect to our topic, we prefer to leave it as an appendix figure.

*The color bars in Figure A1 need to be changed to divergent color bars so the zero value is always white (or gray) for example and positive values are different shades of one color and negative values are different shades of another color, for example blues white reds.*

We used a divergent color scale. It has been centred to meet the request of the reviewer.

*The titles for subfigures d and e need to be changed to Angular speed in 15S-5S (degree/day)*

*The titles for subfigures f ang g need to be changed to Angular speed in 25S-15S (degree/day)*

Done as suggested

*P2L28 after "positive everywhere except a narrow region near 27km over the equator" add "(Figure A1-b)".*

*P2L28 change to: These conditions are stable during the whole **January-March** period.*

*P2L29 In April-June, the rotation weakens and changes sign **(Fig A1-d and A1-f)** while the warming turns to cooling **(Fig A1-e and A1-f)***

Done as suggested

*P2L30 .. the **aerosol** plume stays mostly …. (or something similar)*

Done as suggested

*P2L32 The sentence "By mid-February, the plume has already spread all around the Earth." is clearly wrong, the plume as the authors described in the abstract stays mostly confined between 35S and 20N, that is, it has not spread all around the Earth, I think the authors meant "By mid-February, the aerosol plume has already spread through all longitudes." In any case, there is no evidence of this, so I suggest either adding a figure showing the longitudinal spread or not mention it in this part of the text.*

We mean of course a longitude spread and we argue that "around the Earth" is usually understood in this way in both English and French. The title of the Jules Vernes novel "Le tour du monde en 80 jours" is translated as "Around the World in 80 days" However, the sentence has been modified as suggested. We also refer here to Khaykin et al., 2022 who make a detailed study of the initial dispersion and to our figure 5 which displays such evidence. This is also in agreement with Fig. A1: at the maximal rotation speed of 30 degree per day, the world is travelled in 12 days.

*P2L32. Delete However.*

*P2L34 and reaches and **early** April maximum*

Done as suggested

*P2L35 The sentence "This suggests particle growth" is ambiguous. Do the authors mean, that the OMPS and CALIOP result suggest particle growth or do they mean, that the CALIOP results suggest particle growth. It it is the latter, change the sentence to "Meanwhile, the Cloud-Aerosol Lidar with Orthogonal Polarisation (CALIOP) scattering ratio decreases, which suggests particle growth."*

As CALIOP measures at 532 nm and OMPS measures at 745 nm, the opposite trends suggest particle growth and is shown quantitatively in Fig. 2F & g. We follow the suggestion.

*P2L38 The citation Gorkavyi et al 2021 is only for OMPS. The sentence could be change to something like: The limb instruments suggest larger vertical plume extension than CALIOP due to their coarser vertical and horizontal resolution (i.e., Schwartz et al 2020, Gorkavyi et al., 2021).*

It is not only a matter of the resolution but also of the viewing geometry of limb instruments that generates the arch effect discussed in Gorkavyi et al., 2021. This depends on the curvature of the Earth and applies to all limb instruments in the same way. We do not see that Schwartz et al 2020 discuss this issue besides the general information about MLS. In the case of the plume formed from the Australian forest fires of 2019-2020, the comparison of MLS and CALIOP resolution is betted appreciated in Khaykin et al., 2020. We have changed into "effects of the Earth curvature in the viewing geometry". Such effects make difficult to see the bottom of the plume using limb instruments. We have quoted Schwartz et al 2020 in the Appendix that describes MLS

*P2L40, either say Two separate descent regimes are identified from observations. For the first **regime** …. Or say Two separate descent **phases** are identified from observations. For the first phase*

*P2L46 In the second phase **(after mid-February?)** …*

*Note that the dates in this discussion seems to be based in the results for the 15S-5S latitude band. For the 25S-15S latitude band the regimes are actually separated on March 1st. The authors need to specify which latitude band are they drawing their conclusions.*

It is very difficult to define a precise date, with a one-day accuracy, for the transition between the two phases and even more to see how it varies with latitude. Therefore we give 20 February as a date about which the transition occurs and we choose the word "phase" over "regime". The transition is indicated by vertical bars on figure 2 as recommended below. We also write that "The descent of aerosols separates in two subsequent phases" which means it is not necessary to mention that the second phase follows the first.

*P3L54, if the initial growth of the particles was by hydration until mid march, why is the clear separation of the two regimes in February 15 (for 15S-5S) or march 1 (for 25S-15S)*

The separation of the two descent phases is attributed to the role of the water vapour cooling in the first phase. This effect is bounded by the dilution and by the lower effectiveness of water vapour cooling as the plume descend s towards the neutral radiative level (see Sellitto et al., 2022). Hydration is another process which is also bounded by dilution but does not need to follow the same curve as the radiative cooling. Therefore the progress of hydration until March or April does not contradict the separation of the two descent phases.

*P3L58 Fig 2e does not show any plume information thus, the sentence "The extinction-to-backscatter ratio is also smaller on the periphery of the plume (Fig. 2e)." is not clear.*

Fig. 2e should have been Fig. 2g where the average over the band 30°S-20°N encompassing the periphery of the aerosol plume is shown. This has been corrected.

*P3L60 change to: The similarity of extinction-to-backscatter **(Fig 2g)***

*P3L60 Figure 2g shows data for 3 latitude slices, hence the sentence "The similarity of the extinction-to-backscatter in those two latitude slices" is ambiguous, please modify it.*

The sentence has been modified.

*There is no discussion of the two humps in the vertical motion infer using the water vapor data (Fig 2b), any guesses? Could these be an artifacts of the methodology? How accurate are they?*

These bumps in the initial version are not meaningful and we were not happy with that. They are due to the low vertical resolution of the MLS data that generates jumps in the altitude of the mean. We initially used a 1-km smoothed discretization of the MLS data but this is not enough. We use now a 100 m smoothed version using the non oscillating Akima interpolator before applying the ascent rate estimation. Several values of the time filter have been tried (11, 21, 31 and 41 days). We retain 31 days as the shape of the oscillations does not change with 41 days. For consistency, the same filter is also used now for CALIOP data although not much change is observed with respect with to the 21-day filter. As a result, the bumps disappear and the main result that is the proximity of the water vapour ascent rate to the ERA5 ascent rate is preserved and improved. The two curves still oscillate one around the other and we cannot probably do better here as we compare two zonal means with fairly different sampling.

*P3L65 If I remember correctly Carr et al (2022) only talks about the ashes, there is no mention of ice in that study, please clarify.*

Carr et al. (2022) cannot distinguish ash from ice as they use a single infrared band of the geostationary imagers near 11.2 µm. Nevertheless the result shown in their Fig. 3 is consistent with what we see using the RGB Ash product. In any case thick ice and ash can hardly be separated with broadband channels. We have also added the reference to Khaykin et al. 2022 who discuss this further and demonstrate ice saturation at the altitude of the umbrella cloud on the day of the eruption.

*P3L66. Please change to: What emerges on the west side are two greenish clouds **(Fig 3d)** without any hint of ash **(ashes would appear as yellow/reddish)***

Done as suggested

*P3L67 change The early CALIOP section -> The CALIOP cross section through these clouds show high scattering patches …*

Done as suggested

*P3L68 LOAC is not defined.*

It has been defined

*P3L71 Please clarify how Figure 3g and h shows that the conversion to sulphates started immediately after the eruption.*

We see the sulphates formed with an infrared AOD of about 0.1 just one day after the eruption. That is quite a sign. Furthermore, there is more sulphate where there is less SO2. We have added "with an SA/OD reaching 0.1 one day after the eruption" to make the point.

*P3L75 I think this sentence will be better of changed to: Four days later (Fig. 4b), the two clouds are still separated but …*

Done as suggested

*P3L77 highest cloud? There is no information of height on these plots. Do the authors mean further to the north?*

The two strips have been produced from the two clouds in Fig.3e. This is now clarified in the text.

*P3L80 How was the altitude determined? Please explain in the text.*

This can be inferred from the low angular speed of the cloud at such latitude (see Fig. A1). This is also clarified.

*P4L88 Which supplement movie B? Do you mean the movie described in appendix B. The reader should need to see a movie to understand the paper. Please include snapshots to reach the conclusions discussed in this paragraph.*

The animation is not strictly necessary and Fig. 5 is enough. We recommend however to look at the animation as it is the best way in our opinion to appreciate the dynamics of the patches, much better that a series of snapshot unless it is done on the daily basis which is clearly not possible in a letter. It does not appear so eccentric in 2022 to provide a movie in support of a research paper and most journals recognize this possibility. We provide a mp4 animation which allows image per image viewing with most readers (unlike GIF animations). If the paper is accepted, the supplement movie will be archived and available in the same way as the text.

*P4L84 "The sulphates persist for several months". This could be interpreted as if they have already return to background levels. Please change*

This has been corrected. We now say they persist at least 6 months (from Fig.1).

*P4L92 I don't see patches in c and d do the authors mean Figure 5a and b shows a number of localized concentrated patches …*

*If that is the case please mark the patches with circles or a contour of a distinct color. In particular the patches studied in Figure 6.*

Perhaps we are too used to look at these images but we clearly see patches of higher concentration with oval shapes in panels c and d, albeit diluted with respect to panels. The colour contrast on a printed version depends on the printer and might not as good as on a screen. As said above, the best way to track patches is looking at the animation

Except for 28 January, the cases shown in Fig. 6 correspond to other days than Fig. 5. Figure 5 cases were chosen to show a number of patches that were not covered or split by the blank bands where observations are missing (the product is made from only one of the IASI instruments) Figure 6 cases have been selected from some of the best matches between IASI orbits and CALIOP, and as illustrations of specific features. It happens often that CALIOP misses the patches, makes an overpass over the blank domain of IASI or that the matches are too distant in time. We have many other cases that will be exploited and presented in further studies.

*L95 due this western strip refers to the strips discussed in the previous section if it is please make it clearer.*

Yes it is, this has been clarified.

*P4L96 please add a patch near 11E and* **XXS** *(so that the reader identifies the patch easily).*

Done as suggested. However the patch is that crossed by the two orbit tracks on Fig. 5a, as indicated in the caption of Fig. 6, so it was already well indicated.

*P4L96 same question as before "the eastern strip" refers to the previous section? If yes please make it clearer if not please elaborate or add arrows to the figure.*

Yes it is again. It is not possible to attribute all the patches to one of the two original strips at this stage in the IMS images. There is a significant amount of overlap due to differential rotation and mixing. We would have put marks on the figures if it was that easy. We leave for a further study a more detailed description of the interaction between the two strips and the observed physical properties.

*P4L97 how can we be sure of "along an arc of same angular speed (Fig A1a)" if Fig A1a is an average over 3 months are Figure 6 are snapshots in time.*

We cannot be 100% sure but the stratospheric circulation near 25 km has been boringly zonal during the first month and half after the eruption and we have a long sequence of CALIOP cross sections similar to Fig. 6c. Hence we are almost sure that the explanation in terms of the zonal mean flow is relevant. Again the letter format does not allow to expand all the arguments.

*P4L100. What is a hairy pattern*

This is an analogy for the filamentary structure above the head of the "jelly fish". Perhaps it was not a good idea. It has been changed into "filamentary halo".

*P5L120 Not all around the Earth but rather "dispersed the plume through all longitudes".*

Corrected as suggested.

*P5L120 how many weeks is a few weeks? 2,3, 10. Please be specific.    Further, this has not been shown.*

Less than a month. It is visible in Figure 5.

*P5L122 "about 2km thick"    This has not been discussed (shown) previously please specify on which plot are you basing this conclusion.*

This is easily seen from CALIOP measurements in Figs 1 & 2. We refer to these two figures.

*P5L126 "Our estimation of fall speed and extinction-to-backscatter ratio trends is consistent with a growth up to 1.5-2.0 µm and then a decrease in mean size". This is only after the unrealistic values. Please also mention those here.*

There is a misunderstanding here. It is readily explained in the text that the radius calculation should not be considered during the fall speed due to the water vapour cooling that induces a similar descent rate of both water vapour and aerosols. This does not generate per se a doubt on the estimates made during the second phase. It is not unrealistic, it is just invalid. Perhaps, the confusion was due to the word "discrepancy" that we used on p2L43. There is no discrepancy if we explain the observed behaviour. This has been modified.

*P5L129 delete strongly. There is no calculation in this study to suggest a strongly underestimation, there is only evidence of an underestimation but not of its magnitude.*

Corrected

*P5L139 add i.e., before the Solomon citation (other studies have shown the potential of water vapor as well)*

Perhaps the referee means e.g.

*P5L139 add Millan et al 2022 to the S2022 reference.*

Millan et al. 2022 make a general comment that H20 cooling can increase the Brewer-Dobson circulation but do not consider at all the possibility on an intense cooling within the concentrated plume that can induce a fast descent of the plume. They comment about the TOA and surface radiative forcing by quoting S2022. We consider the descent due to cooling as an original contribution of S2022. It is not mentioned either in Schoeberl et al 2022 who nevertheless correctly describe and interpret the second phase of the separation between aerosols and water vapour.

*P6L161. Why are you using the near real time data? there should be a more accurate product. Near real time data are normally only used for situations where near real time data is needed. The data discuss in this study is from 8 months ago.*

The IMS product is provided in near real time like a number of other satellite products which have applications in numerical weather and air quality forecasts. There is so far no other version for this product.

*P6L165. Do you mean SO2?*

Yes, this was a LaTeX error.

*P6L175. How was the conversion from pressure to altitude done for MLS data? Please specify in the manuscript.*

We use the ERA5 data for this conversion. The conversion is performed daily in the zonal average framework. In the considered altitude range, the levels of the ECMWF model are pressure levels.  This is now mentioned in the appendix.

*P8L218. This details about the python based code should be in the code availability section. Here just say: The theoretical extinction-to-backscatter ratio for the plume has been calculated using Mie calculations. The extinction and backscatter coefficients have been …*

We think it is important to draw the attention of the reader who might be interested in conducting such calculations as this tool is very useful and handy.

*Figure 2*

*For panels b, d, and e,   please add a vertical line indicating the separation of the two regimes. In the caption you could say something like: The vertical line separates the two regimes (or phases) as discussed in section X.*

Done as suggested.

*The title for panel a and c need to be changed to  "… band 15S-5S" and to ". band 25S-15S"*

*The title for panel b need to be changed to Vertical motion in 15S-5S*

*The title for panel d need to be changed to Vertical motion in 25S-15S*

*The labels in e need to be changed to 15S-5S and 25S-15S (similarly the labels in g need to be changed)*

Modified as suggested

*Figure 3*

*Please add the location of the volcano in panels a-d.*

Done as suggested.

*Please repeat the color information in the caption. That is, in the caption include This product allows to qualitatively distinguish thick ash plumes or ice clouds (brown), thin ice clouds (dark blue) and sulphur-containing plumes (green). Mixed ash/sulphur-containing volcanic species would appear in reddish and yellow shades.*

Done as suggested.

*The (g-h) caption is wrong. It should say 16/01/2022 instead of 16/02/2022.*

Corrected

*Also, I strongly recommend using the same format for all dates through-out the study.   For example, use the format 28 Jan 2022 as used in Figure 1.*

Done as suggested

*Figure 4 The color bars for SA OD through out the sub figures should match so the reader could easily intercompare the values. The same for the SO2 colorbars.*

*That said, all SO2 plots and all SA OD should have the same color bar ranges so that the reader could easily intercompare them.*

We have homogenized the color scales but there is no reason to use the same one in an instantaneous lat x lon chart and in a zonal mean. We need also to accommodate the dilution with time which is particularly strong for SO2. Nevertheless, the color scale is common in all panels of Fig. 5 and we use always the same minimum value. The color scale of Fig. 5 is used in two panels of Fig.6, a different maximum being used in a third panel to better exhibit the tripolar structure.

*Figure 5   Either the D and N labels are wrong ore the caption is wrong. Please double check and fix.*

The caption has been corrected.

*Figure 6 All other figures al labeled (a,b,c,d,etc) from left to right and top to bottom. Please change the labels to be consistent.*

The figure has been reshaped into a vertical two-column format that improves the reading and complies to the standard labelling convention.

*Figure A2. Add latitude label. The color bar for this figure should also use a divergent colorbar.*

Done as suggested

References

- Carr, J. L., Horváth, Á., Wu, D. L., and Friberg, M. D.: Stereo Plume Height and Motion Retrievals for the Record-Setting Hunga Tonga-Hunga Ha'apai Eruption of 15 January 2022, Geophysical Research Letters, https://doi.org/10.1029/2022GL098131, 2022.

- Gorkavyi, N., Krotkov, N., Li, C., Lait, L., Colarco, P., Carn, S., DeLand, M., Newman, P., Schoeberl, M., Taha, G., Torres, O., Vasilkov, A., and Joiner, J.: Tracking aerosols and SO2 clouds from the Raikoke eruption: 3D view from satellite observations, Atmos. Meas. Tech., 14, 7545–7563, https://doi.org/10.5194/amt-14-7545-2021, 2021.

- Khaykin, S., Legras, B., Bucci, S., Sellitto, P., Isaksen, L., Tencé, F., Bekki, S., Bourassa, A., Rieger, L., Zawada, D., Jumelet, J., and Godin-Beekmann, S.: The 2019/20 Australian wildfires generated a persistent smoke-charged vortex rising up to 35 km altitude, Commun Earth Environ, 1, 22, https://doi.org/10.1038/s43247-020-00022-5, 2020

- Khaykin, S., Podglajen, A., Ploeger, F., Grooß, J.-U., Tence, F., Bekki, S., Khlopenkov, K., Bedka, K., Rieger, L., Baron, A., Godin-Beekmann, S., Legras, B., Sellitto, P., Sakai, T., Barnes, J., Uchino, O., Morino, I., Nagai, T., Wing, R., Baumgarten, G., Gerding, M., Duflot, V., Payen, G., Jumelet, J., Querel, R., Liley, B., Bourassa, A., Hauchecorne, A., Ravetta, F., Clouser, B., and Feofilov, A.: Global perturbation of stratospheric water and aerosol burden by Hunga eruption, preprint, https://doi.org/10.1002/essoar.10511923.1, 20 July 2022.

- Millán, L., Santee, M. L., Lambert, A., Livesey, N. J., Werner, F., Schwartz, M. J., Pumphrey, H. C., Manney, G. L., Wang, Y., Su, H., Wu, L., Read, W. G., and Froidevaux, L.: The Hunga Tonga-Hunga Ha'apai Hydration of the Stratosphere, Geophysical Research Letters, 49, e2022GL099381, https://doi.org/10.1029/2022GL099381, 2022.

- Podglajen, A., Le Pichon, A., Garcia, R. F., Gerier, S., Millet, C., Bedka, K. M., Khlopenkov, K. V., Khaykin, S. M., and Hertzog, A.: Balloon-borne observations of acoustic-gravity wavesfrom the 2022 Hunga Tonga eruption in thestratosphere, preprint, https://doi.org/10.1002/essoar.10511570.1, 13 June 2022.

- Proud, S. R., Prata, A., and Schmauss, S.: The January 2022 eruption of Hunga Tonga-Hunga Ha'apai volcano reached the mesosphere, preprint, https://doi.org/10.1002/essoar.10511092.1, 17 April 2022.

- Schoeberl, M., Ueyama, R., Taha, G., Jensen, E., and Yu, W.: Analysis and impact of the Hunga Tonga-Hunga Ha'apai Stratospheric Water Vapor Plume, preprint, https://doi.org/10.1002/essoar.10511762.1, 5 July 2022.

- Schwartz, M. J., Santee, M. L., Pumphrey, H. C., Manney, G. L., Lambert, A., Livesey, N. J., Millán, L., Neu, J. L., Read, W. G., and Werner, F.: Australian New Year's PyroCb Impact on Stratospheric Composition, Geophys. Res. Lett., https://doi.org/10.1029/2020GL090831, 2020.

- (S2022) Sellitto, P., Podglajen, A., Belhadji, R., Boichu, M., Carboni, E., Cuesta, J., Duchamp, C., Kloss, C., Siddans, R., Begue, N., Blarel, L., Jegou, F., Khaykin, S., Renard, J.-B., and Legras, B.: The unexpected radiative impact of the Hunga Tonga eruption of January 15th, 2022, preprint https://doi.org/10.21203/rs.3.rs-1562573/v1.

- Taha, G., Loughman, R., Colarco, P. R., Zhu, T., Thomason, L. W., and Jaross, G.: Tracking the 2022 Hunga Tonga-Hunga Ha'apai aerosol cloud in the upper and middle stratosphere using space-based observations, 2022, under review in Geophys. Res. Lett.

- Witze, A.: Why the Tongan eruption will go down in the history of volcanology, Nature, 602, 376–378, https://doi.org/10.1038/d41586-022-00394-y, 2022..

- Zhang, H., Wang, F., Li, J., Duan, Y., Zhu, C., and He, J.: Potential Impact of Tonga Volcano Eruption on Global Mean Surface Air Temperature, J Meteorol Res, 36, 1–5, https://doi.org/10.1007/s13351-022-2013-6, 2022

**Answer to referee 2**

We thank the referee for his/her thorough assessment of our work and the detailed comments. Although the initial comments of the referee need to be replied, we reply tthers only to the comments that do not reapper later among the specific comments.

*This manuscript presents an analysis of the progression of the volcanic aerosol cloud from the January 2022 Hunga-Tonga eruption Other papers in review or recently published have explored other aspects of the Hunga-Tonga event with the Khaykin et al. study analysing the global-mean stratospheric AOD and water vapour from the eruption cloud, illustrating each are the strongest global perturbation in the post-Pinatubo period. But as far as I am aware this is the only paper to present a progression in the vertical profile and meridional extent of the aerosol cloud.*

We are aware of another paper by Schoeberl el al., 2002, submitted by early July who also describe the separation of rising moisture from descending aerosols

*I am conscious I have not been able to complete the review of this manuscript until now (12th August), and I am sorry for not being able to submit this review sooner.*

*In particular, I noticed that, in the period since this manuscript was submitted, another manuscript analysing the aerosol and water vapour within the volcanic cloud from the Hunga-Tonga eruption cloud (Khaykin et al., 2022) was submitted (on 31st July).*

*I understand that the authors must be keen for their results to appear as soon as possible, and the choice to submit a short ACP letter may have been partly motivated by an aim to achieve publication on a shorter timescale than would be the case for a regular ACP article.*

*So I apologise again for not being able to submit this review before now.*

*I am also aware another manuscript led by one of the co-authors of this study (Sellitto et al., 2022) has been submitted to another journal, which assesses the radiative effects from both the aerosol and the water vapour. That manuscript explains the unexpected strength of the radiative effects, both in terms of the aerosol forcing being higher than the modest 0.4Tg of SO2 measured to have been present, but also in terms of the surface warming effect from the >100 Tg of water vapour leading to the first well-observed case of an overall net warming eruption.*

This paper is indeed complimentary to our work and we mention also Khaykin et al, 2022 that focuses more on the water vapour transport.

*This Legras et al. manuscript has an "applied for MS type" set for "ACP Letter", and see that ACP has this type requiring an enhanced criteria of "particularly important results and major advances":*

*https://www.atmospheric-chemistry-and-physics.net/about/manuscript_types.html*

*Whereas the main focus of the Sellitto et al. study is in relation to the radiative effects, the present study, submitted for ACP letter is the first to present a multi-platform assessment of the longer timescale progression of the scattering-magnitude of the aerosol cloud, through to the end of May 2022, and to assess its progressing altitude and depth, Figures 2 presenting for example the descent rate within the initial weeks after, and the later phase showing slower descent.*

*The enhanced criteria for an ACP letter are also clearly achieved, with these results being important to understand how the aerosol cloud is continuing to progress, in relation also to potential impacts of the emitted water vapour and may still have on the ozone layer, and either the latter phase of this year's Antarctic ozone hole or potentially for next season.*

*One key aspect in this regard is the fact that Figure 2 illustrates that the 15-25S latitude band has a continuing unprecedented >10 ppmv water vapour concentration at ~24-28km, this very-strongly-enhanced level apparent*

*throughout March, April and May. And furthermore whereas the lack of variation in the 15-25 South band shown in Figure 2c is remarkable.*

*Furthermore, the latter results from Figure 2c (from the first weeks of June) give (for the first time in published article I expect) an indication as to for how long the unprecedented high water vapour concentrations may continue for, with there being a first indication that the water vapour is now beginning to reduce.*

*In my specific comments (comment 4) I am suggesting the authors extend the x-axis for Figure 2 to include the results through to end of July (or even into the first week or so of August, to then be able to give a clearer indication of whether indeed there is a continuing decreasing signal from the extreme 10 ppmv values the eruption has caused.*

Figure 2 has been extended to 26 July. The situation evolves very slowly in the following weeks and we have a two-week gap in the OMPS data after 26 July that would truncate some of the curves.

*The manuscript's focus is on the aerosol portion of the cloud, with this important complimentary finding from the latest results re: the longevity of the high water vapour concentrations in the Southern Hemisphere mid-latitude stratosphere caused by the eruption, I am requesting in that comment 2 that the authors add a sentence to the Abstract, to provide this prominently within the Abstract after the aerosol results have first been summarised*

*Although the Figures illustrate very well the main results from the study, some parts of the text in the manuscript require improvement before the manuscript can proceed to publication as an ACP letter.*

*And I have provided below a list of specific suggested minor revisions which are mainly seeking to improve the wording where the main results are summarised in the Abstract and conclusions.*

*I am conscious the authors are not native English speakers, and although there are a large number of suggested edits, these are mostly minor in nature.*

*Only 2 of the specific revisions might be considered major, the first being the request to indicate within the title the main focus on the manuscript being re: the aerosol properties, with suggestion to add text in the Abstract mentioning the slow but clear separation of the aerosol from the water vapour, 2 distinct plumes emerging during March and April (evident from Figures 1 and 2).*

*Whereas the Millan et al (2022) manuscript analyses the period up to the end of March, this manuscript is the first to analyse this longer timescale progression through April and May where the aerosol has a slow but steady descent, whilst the water vapour plume remains at the same altitude or with slight ascent.*

*Whereas the early-phase of the very high water is documented in the Millan et al. (2022) GRL paper (submitted 30th April, accepted 2nd June), the analysis in that paper extends only to the end of March.*

*The MLS results shown in the present manuscript's Figures 1 and 2 show the more situation in more recent months, and potentially indicate the scenario forward-projection model simulations shown in Zhu et al. (2022) may even have been an underestimate for the amount of water vapour in the winter mid-latitude Southern Hemisphere stratosphere.*

The persistence of moisture is now mentioned in the abstract and commented in the discussion. However, we do not use any modelling use that allows to foresee the future and the long-range impact of the plume, in particular regarding the ozone hole, is beyond our scope.

*The other more-substantial comment is in relation to the interpretation within the impressive analysis to assess the descent of the aerosol portion of the cloud in the initial weeks after the eruption, or the portion that dominated the backscatter signal at that particular time.*

*The approach is novel and interesting, enabling to make quantitative statements about the progression of the aerosol cloud, potentially revealing that there has been a systematic removal of some sub-population of the aerosol with differing characteristics to the others. This could potentially be as a results of particles at larger sizes*

*sedimenting faster (such as a more-strongly-hydrated fraction, ash particles, or even potentially volanically-detrained marine aerosol).*

*Whilst this analysis is another excellent part of the analysis, using the term "aerosol motion" in the methodology section (line 215, Appendix A3) and also subsequently in the text, is too specific an attribution, in my opinion. The further analysis to derive an "aerosol radius" in Figure 2e is certainly too strong, and in the next para I request the authors revise this size-association to a be clear this is an "apparent size" or similar. The text also needs to be moderated to be consistent with this being indicative of a particle size, also in relation to the sizes dominating the backscatter signal at the wavelength observed.*

*My specific suggestion is to change these two terms instead to "apparent aerosol descent rate" or similar and "optically-derived aerosol radius" or similar. The term "effective radius" has a specific translation to the ratio of volume to surface area, so that term should not be used.*

*I recommend the term "apparent descent rate" is used, then communicating implicitly this is a derived quantity, the "optically-derived" being required for the derived-size, to remind that there may well be other smaller aerosol sub-populations present, that descend more slowly, considering that smaller-sized particles tend to be under-represented within the optical signal measured from the lidar detector.*

*The Figure 2b) and 2d) also label the descent of the CALIOP-derived mid-visible extinction as "vertical motion", but so certain a translation from the measured optical properties, although appropriate for the descent of that signal, I'm advising not be denote with the word "motion".*

*Suggest to change "Vertical motion" instead to "apparent descent rate" or similar. Whereas that is a more tentative suggestion that I leave it to the authors to decide, the further analysis to associate an "aerosol radius" from the apparent descent rate the request to change "aerosol radius" to "apparent particle size" fall speed the aerosol descent rate translates into, where I am requesting some changes in the interpretation.*

*These 2 more substantial changes are straightforward to implement however, and can be considered minor in character, and my overall assesment then is to recommend the manuscript be published as an ACP letter, once the set of specific revisions below have been made.*

This is a main point which deserves a fairly detailed reply as we have some difficulties to understand the concern of the referee and we are reluctant about the systematic usage of the word "apparent" suggesting that our interpretation is, a priori, invalid. Most information on atmospheric aerosols is from optical measurements and inversion by various instruments and methods. There is a huge specialized literature that validates these measurements and provide confidence in the results albeit the uncertainty remains quite large in practice. In the case of this paper, we do not use any inversion but rely on the physically measured quantities, backscatter or extinction that can be directly related through the calibration of the instruments to the photon count of the sensors. Therefore we consider as granted that we see an aerosol layer and not an apparent aerosol layer and that its location is well determined from the CALIOP lidar as this is just a matter of time counting, something that can be done with high accuracy. That this layer exhibits a vertical descending motion cannot be disputed. It does not necessarily represent the whole of the aerosols. There might be other undetected much smaller aerosols but these latter will not fall, will be entrained in the Brewer Dobson upwelling and likely evaporate as noticed by Schoeberl et al. 2022. Anyway, we can only discuss and interpret what we see, not what we miss. The simplest interpretation is that we see falling particles and it is then reasonable to estimate a radius from this falling speed and see whether this is consistent with the other pieces of the puzzle. The main other piece is the "lidar ratio" calculated from OMPS-LP and CALIOP. The theory (Fig.2f) says that this ratio increases when the particle size increases and decreases when the particle size decreases. This is also what we find in the data (Fig.2g). We also see a first phase of the evolution where both the water vapour and the aerosols descend very fast and at the same speed. We explain it from the radiative cooling of the water vapour which is calculated in Sellitto et al. (2022).

The referee makes two specific reservations about the presence of ashes and the possibility of aerosols of marine origin. The significant presence of ash is not supported as there is no depolarization, no detection by UV instruments or in situ spectrometer, and evidence that the ash and ice cloud sedimented very fast in the first hours following the eruption. We discuss below arguments that no not support the marine aerosol hypothesis. The fact that the aerosol plume thickness stays almost constant with time after the first month (see Figs 1 and 2) calls for a fairly compact size distribution of the scattering aerosols. Therefore, we do not see any support for the specific reservations made by the referee and we cannot make reservation, based on Popper's principle, on such general statements as " *there has been a systematic removal of some sub-population of the aerosol with differing characteristics to the others".* If the referee has some specific support for his/her hypothesis, they need to be made available to us or published before we can discuss them.

We write "the apparent aerosol radius is estimated by interpreting the aerosol plume motion as a fall speed of the scattering particles using Eq. 9.42 of Seinfeld and Pandis, 2016." and we think that there is no need to associate the word "apparent" to each occurrence of the word radius.

*List of specific revisions.*
* * *
*1) Title -- Re: the first of the two potentially-major comments, and my summary statements above, although I'm conscious it's possible other manuscripts are in review that I am unaware of, to my knowledge this manuscript is the first to present such a comprehensive assessment of the optical properties of the aerosol portion of the Hunga-Tonga volcanic cloud.*

*In my general comments above, and in specific revision 4) below, I'm suggesting to update the Figure 2 timeseries to complete through to the end of July, which would then have the first 6 months after the eruption. And then although the water vapour is not the main focus of the article, adding specific mention of the 6-month timescale that the article assesses will be of benefit to readers being able to refer to this paper also for this issue being able to assess the longevity of the unprecedented high water vapour concentrations, and even see first indications of a potential decline.*

*Given there will be interest in the water, my specific suggestion is to consider adding a new 2nd part of the title:*

*"and the progressing aerosol properties in the first 6 months post-eruption"*

*I wonder if there could even be the potential to refer to the result re: the separation from the water vapour portion of the plume, with a longer 2nd part, continuing after the current "in the stratosphere" with a colon ":" and then*

*": progressing aerosol properties and vertical separation from the water vapour in the*

*first 6 months post-eruption".*

We are, as a general rule, more in favour of short titles, easy to catch the attention of the reader and to memorize, even if they cannot be fully descriptive, but we see the point of the referee and we have changed the title in the suggested way.

*2) Line 1, Abstract -- suggest to add 1 sentence re: the continuing high water, and also a 2nd extra sentence re: the distinct vertical-separation that has emerged in recent months between the altitude of the volcanic aerosol and the altitude of the volcanic water vapour.*

*This aligns with the suggestion above to add specific mention in the title, and then the Abstract can 1 sentence stating this finding re: the continuing high water beginning to decline in June (and July if that turns out to the case when the Figure is updated).*

*I have provided a suggested wording this:*

*"We also show the unprecendented high water vapour concentrations shown in the MLS measuremnts in February and March, are in this study shown to have continued throughout April and May, a > 10 ppnv in the altitude range 24-28km at 15-25S."*

*And in the 2nd sentence to add*

*"We also show that in these recent months, there has been a distinct "vertical separation" has progressed with the aerosol now at 22-24km, the water vapour remaining at ~25-28km altitude."*

*The Abstract is relatively short in the submitted manuscript and I feel there is definitely space for the 2 additional sentences actually.*

Actually we do not have that much freedom as the submitted abstract fills exactly the ACP Letters 200 words limit  There is a sentence saying "As sulphate particles grew through hydration and coagulation, they sediment and separate  from the ascending moisture entrained in the Brewer-Dobson circulation." which accounts for the information required by the referee. We have updated the sentence describing the duration of the plume.

*There could even be an opportunity to add a 3rd sentence: there being an apparent decline, and in relation to the impacts on the Antarctic ozone hole in the latter part of the 2022 season or whether high water vapour could reach the 2023 Antarctic polar vortex when it spins up in April 2023.*

*A suggested wording for this potential 3rd extra sentence is:*

*"We also see first indications for the rate of decline from these high concentrations, important given the potential for a continuing high water vapour shown to affect the Antarctic ozone hole."*

This is an interesting topic worth of investigation but we decided to stay away from it in this paper due to the format restrictions and because we cannot produce more than a speculation. In addition we do not think the abstract is the right place for a speculation that is not supported by any specific result. In the revised version we keep showing data in the 35S-20N range event if aerosols and water vapour have been obviously transported to lower latitudes in late spring and summer as the wave activity increased in the southern stratosphere. The issues related to water vapour are receiving a more detailed attention in a separate paper by Khaykin et al., (2022), with a number of common authors, which has been submitted after this one and is now quoted  in the text

*It is obviously a decision for the authors to make, to determine whether to add these sentences, but I think the paper's status as an ACP letter will be elevated with this broader context for the results presented.*

*t readers to the observational results presenting this indication of the continuing unprecedented high water vapour in the Southern Hemisphere stratosphere, I suggest the authors add an extra sentence to the Abstract flagging up this result.*

*I think the 2 or 3 suggested extra sentences would work as an extra final part of the Abstract, or be incorporated somewhere earlier within a longer 2nd half of the Abstract.*

*I am conscious that the ACP letter format requires only a short Abstract, and I am not sure whether the current Abstract is already close to the maximum allowed word limit.*

*But given the importance of the results, I wonder if the Editor could potentially give special permission for a modest extension to the reduced form usually required for an ACP letter.  The article itself will still conform to the required length, so a minor adjustment to enable this information to be provided in the Abstract would be of benefit to the journal, in terms of potentially raising awareness of there being this "ACP letter" manuscript format option when submitting to ACP.*

We have been told that we should not take too much freedom from the rules of ACP letters even if we hope we will be permitted to do the minimal changes required to meet the legitimate requests of the referees.

*3) Abstract line 3 -- The authors state with absolute certainty the initial plume was "without ashes", but on lines 64-65 refer to "the ash and ice plume". Whilst the focus of this article very usefully assesses the progressing composition of the plume, the statements need to be more nuanced to be clear the authors are arguing the ash was removed within initial days.*

*There was clearly injection of ash in the stratosphere but nothing suggests that it remains visible in the stratosphere after the first day but perhaps for a thin cloud at 35-40 km that is described in Kha*

*Also, the term "washed-out" is not correct, because the term "washout" refers to below-cloud scavenging of aerosol from precipitating rain drops, the term then not appropriate for the stratosphere.*

*The authors say "within the first hours", but whilst the very strong depolarisation seen in the 16th January CALIOP profiles are not seen after that time, there are moderate depolarisation seen in later CALIOP profiles which indicate at least some parts of the plume may still have ash, and the wording needs to be more precise here.*

The only strong depolarisation seen on 16[th] January, about 50%, is on the thin cloud seen at 35-40 km, moving westward with a mean speed of about 40 m/s and submitted to a large shear. This cloud was seen a few days later from a lidar at La Reunion (A. Baron, personal communication, Khaykin et al, 2022) and then lost, at least from CALIOP data which show no trace after the observations resume on 27 January. We omitted these results due to space limitation and because they are described in Khaykin et al. 2022. This cloud is too thin to be seen from Himawari imager, unlike the initial plume just after the eruption, and is also totally separated from the two thick components of the plume which are described in this paper. The depolarisation remains always under 1% during the whole history of the aerosol plume, a value which is usually not considered as moderate but low. Larger values are only observed as noise where there is no aerosol. Although the presence of very thin ashes cannot be totally excluded, for instance inside the aerosol droplets, it does not show up optically and thus remains purely speculative.

*I am not sure whether the authors are indicating the ash may have been encased within ice, and then removed by sedimentation more rapidly than in other cases, but the wording needs to be made more precise, and if they are advancing this suggestion that the ice sedimentation may have potentially provided an enhanced removal mechanism the ash case, the wording should at hint at this explicitly.*

*I am not sure if the "washed-out" and "washed-down" is actually the authors arguing the ice-sedimentation removal pathway, but if so the wording could be adjusted to hint this.*

*Please change "washed-out within the first hours" to "removed within the first days", and suggest to potentially add "possibly via sedimention within larger ice particles" or similar wording.*

The circumstances of this eruption are very unusual. The GPS-RO profiles in the close vicinity of the plume on the day of the eruption suggest that just after the eruption a saturated profile of water vapour was established up to the top of the umbrella as shown in Khaykin et al., 2022. Therefore it is very likely that all the water in excess of the saturation has condensed in big ice crystals and has scavenged the ashes in the same way as precipitating rain does usually in a low cloud. We do not see any other mechanism that can explain the sudden collapse of the ash and ice cloud which was very fast compared to other recent well documented eruptions such as the Raikoke in 2019. Same analysis was made by Taha et al. (2022, in revision for GRL) who additionally mention that TROPOMI UV data show SO2 but no ash on 17 January. Therefore, we keep "washed-out" in the abstract for the sake of compactness and we change the text sentence into "removed within the first day after the eruption likely via sedimentation within large ice particles". This is much more than a possibility.

*4) Figure 2, line 14 -- as in the above comments, suggest to extend the x-axis of this Figure to continue through to end of July, to then be able to provide the full first 6 months after the eruption, and the latest information re: the indication from the early-June results in Figure 2c) of a potential decline in the unprecedented high water vapour of >10 ppmv measured by MLS.*

The axis has been extended until 26 July. We gathered data after this date but there is little change over the following weeks and since OMPS data are not available between 26 July and 12 August, this would have interrupted several of our data series and plots.

*5) Abstract, line 5 -- The word "While" is confusing in this context, I think the authors mean "Whilst". But suggest better to re-word "While SO2 returned to" instead to*

*"Whereas SO2 had returned to...". Or alternatively change "While" to "Whilst.*

Done as suggested.

*6) Abstract, lines 8-9 -- This sentence beginning "Sulphate aerosol optical depths" needs to be clear what timescale the organising into concentrated patches is referring to.*

*Do the authors mean the initial days after the eruption -- please add specific mention of the timescale here -- it's only for the first few days or weeks this occurs, right?*

The timescale is the first two months and the abstract has been corrected to mention it.

*I also wonder if the authors are indicating some relationship between the strong radiative cooling from the emitted water vapour and the dynamical structures observed? Are these unusual compared to what has been observed for other eruptions?*

*I'm wondering if the unprecedented altitude at which the Hunga-Tonga eruptive plume detrained might unusually have preserved these structures when usually the closer proximity to the tropopause would see the plume dispersion more disrupted, the structures then not so apparent?*

*Please add a few words to this sentence of this Abstract to mention both the timescale the structures remain, and any specifics in relation to the higher altitude detrainment and/or the strong radiative effects from the emitted water vapour.*

The abstract says that AEOLUS data suggests the structures are related to vorticity anomalies and that they are very similar to the product of shear-induced instabilities, that is active rather than passive dynamics. This is all we can say at the moment. The PV gradient shown in Fig. A2 suggests the need of local enhancement of the shear or a local vorticity source. This can be produced by radiative heating and has been shown to be very effective for forest fire plumes, even close to the tropopause (Khaykin et al., 2020; Lestrelin et al. 2021). In the present case, the effect is not detected by the ECMWF analysis or reanalysis and needs further investigation coupling radiative calculation with dynamics. It is indeed possible that water vapour cooling played a role in generating vortical structure during the first two weeks. Meso-vortices are often detected after large volcanic eruptions but are short lived features for a few days only. The only case of long duration structure we are aware of concerning a volcanic eruption is the compact patch followed by Chouza et al. 2020 after the Raikoke eruption which has been surprisingly to our eye interpreted as a passive structure by Gorkavyi et al., 2021.

*7) Introduction, line 14 -- The authors state the "explosive intensity is close to that of Mount Pinatubo in 1991", but the Wright et al. (2022) paper presents evidence the eruption was actually more explosive than Pinatubo, with explosive energy quantified to be comparable to 1883 Krakatau. Suggest to either change "to that of the eruption of Mount Pinatubo in 1991", or if that is a finding from another paper, then to add separately at the end of the sentence reference to the Wright et al. (2022) analysis suggesting on the scale of 1883 Krakatau.*

This is based on the volcanic explosive index estimated by Poli and Shapiro, 2022 from seismic data and a model of the magmatic system. The estimate by Wright et al., 2022, is based on the emitted Lamb wave. The two estimates do not tell necessarily the same story about the eruption and it is not within our scope to reconcile these two approaches. We have modified the sentence about the Krakatau into "The induced atmospheric Lamb wave circled the globe at least 4 times with an amplitude comparable to that of the 1883 Krakatau eruption" and added Wright et al. 2022 to the references.

*8) Introduction, line 19 -- change "the stratosphere" to "the upper stratosphere"*

The second part of the sentence says "increasing its overall water vapour content by 10%" and is meant for the whole stratosphere. It would be misleading to change "the stratosphere" to "the upper stratosphere".

*9) Introduction, line 25 -- Change "mean zonal pattern" to make the wording more specific to the pattern being described.  Also the word "pattern" is a little non-scientific unless part of an analysis systematically applying a pattern-matching algorithm or so.*

*I think the word "zonal" is in relation to the plume's dispersion in the zonal direction, or maybe the authors mean the zonal variation of the plume in the initial days/weeks?*

*Please then change either to "The zonal dispersion of the Hunga-Tonga plume" or "Zonal variations within the Hunga-Tonga plume dispersion" or "Structures within the zonal dispersion of the Hunga-Tonga plume" or similar.*

Thanks for pointing out that the header of this section might be confusing. This section does not deal with the zonal dispersion but describe the large scale evolution of the plume as seen from zonal averaging. It has been renamed "Six-month evolution of the zonal mean". In this work, we decided to proceed from the  global impact view to the details rather than present a series of events in chronological orders.

*10) Introduction, line 31 -- The authors refer to Figure 1 showing an initial "fast latitudinal dispersion" but the extent of that latitudinal spread needs to be clear, and the word "meridional" is clearer than "latitudinal".*

*Suggest to change "fast" to "rapid", and add what timescale is intended here -- "in the first days after the eruption" or similar.*

We made the changes "latitudinal" to "meridional" and fast to "rapid" and we refer to Khaykin et al. 2022 who provide a description of this initial dispersion.

*11) Introduction, line 43 -- The authors explain the water vapout is descending, and note this being in opposite sense to the rising ERA5 motion, but the use of the word "against the rising ERA5 motion".*

*Suggest to simply change "against" to "in contrast to" and change "the rising ERA5 motion" to "the rising motion in ERA5".*

Done as suggested.

*12) Introduction, line 46 -- The authors explain a later second phase where the water vapour (then diluted to lower concentrations) begins to rise, and state this is "in agreement with ERA5 upwelling".  Similarly to the word "against", better to change "in agreement with" to "ascending at the same rate as".*

Done as suggested

*13) Line 63, Section 3 sub-title -- Whilst this section presents an interesting discussion of the composition of the plume, since this is not measured directly, the word "Inferred" needs to be added at the start, the sub-title changed to "Inferred composition of the plume"*

Done as suggested

*14) Line 64, Section 3 -- Re-word "We now consider the history of the aerosol composition of the plume" instead to something more nuanced for the measurements being interpreted, such as "We now consider what the satellite measurements indicate for the early-phase composition of the Hunga-Tonga plume, and some initial cases where we infer the optical properties indicate that a change in the dominant scattering sub-population may have occurred".*

*Further to the comments above, since the aerosol particles are not being sampled, but inference from the optical properties, it's important to communicate to the reader the derived nature of the composition, at least in the initial sentence of the paragraph.*

Most instruments measuring atmospheric composition in gas and aerosols, in situ or remotely, rely on optical properties and on the interpretation of the basic measurements with physics. We think that the reader of ACP is generally aware of that situation and does not need specific warning, especially in a letter format where the words are counted. We also use the in situ LOAC information, still based on optical properties, but it is not a satellite. Therefore we politely disagree here with referee on the need to change the introductory sentence of this section. See also our general comment above about "apparent" results.

*15) Line 65, Section 3 -- Re-word "is rapidly washed down" (line 65) to "has a steeply descending altitude". The signature here may be indicative of another process, such as wind-shear or some dynamical aspect of the eruptive plume's detrainment. Whereas the descent within the longer-timescale variations in Figures 1 and 2 can be more certainly attributed, the daily-timescale vertical-shearing of the plume is clearly apparent within the CALIOP measurements (e.g. Sellitto et al., 2022), and from the ground-based lidar measurements at Reunion Island (e.g. Khaykin et al., 2022).*

*There are a range of possible causes in these initial hours after the eruption, and need to be cautious not to attribute too certainly a variation to a particular process.*

Referring to our reply to comment 3 above, wind shear alone would not produce a vanishing of the initial ash signal on the time scale of a few hours or even a few days, preserving SO2, and we are not aware of any dynamical process that could do the job. Therefore we argue again that the fast fall of large ice+ash particles in the saturated stratosphere is the simplest and most plausible mechanism which is consistent with the observations. If there are other possibilities, they need to be defined and testable.

*16) Line 69, Section 3 -- Re-word "hence made of" instead to "indicative of predominantly".*

Done as suggested

*17) Line 69, Section 3 (Inferred composition of the plume) -- Re-word "sub-micronic" to "sub-micron sized".*

Done as suggested

*18) Page 13, Figure 1 -- Whilst most expert readers will be aware of the difference between the observed quantity shown in the OMPS, CALIOP and MLS, and that information is provided in the Figure-caption, also considering the difference in wavelength between the OMPS and CALIOP aerosol extinction, suggest to add label on the far-right each row of the Figures to aim to ensure the reader can be mindful of this when comparing the different rows in each column.*

*A specific suggestion is for 3-line label in each case, the OMPS 3 lines being upper-line = "aerosol", with then "extinction-ratio" immediately below that, with then "(745nm)" as the 3rd line. Similarly, for the 2nd and 5th rows of sub-panels have the label at the far-right (for CALIOP) as "aerosol" "backscatter-ratio" and "(532nm)".*

*And then to have "water" "vapour" and "(ppmv)" as the label. Whilst this might seem quite a specific request, it will both help readers see the main result, and also ensure the inference in terms of the specific quantity and units shown is correct.*

We are unsure this is needed as the information is already available twice on the header and the caption but, since it was easy, it has been done as suggested.

*19) Page 13, Figure 1 -- The grid-lines shown in each sub-panel are very useful for the reader, whereas the gold/orange coloured lines are good for the darker background colours in the OMPS and CALIOP sub-panels, the gridlines cannot be seen in the MLS panels when overlaid on the aqua coloured contouring for the background (or the green colours). Please use a darker colour for the gridlines for the MLS Figure sub-panels shown in the 3rd and 6th of the rows of sub-panels in the Figure.*

Done as suggested

*20) Page 14, Figure 2 -- As explained in main comments, in Figures 2b) and 2d) change "vertical motion" to "vertical motion / apparent descent rate" and in Figure 2e) change "Aerosol radius" to "optically-derived aerosol radius", also revising the caption text accordingly.*

See our general comment at the beginning of this reply. The aerosol radius is derived from the fall speed formula that does not involve optical properties and differs from the effective radius which is related to the optical properties. Following the suggestion of the referee would generate confusion.

*21) Page 15, Figures 3 and 4 -- The panel h) of Figure 3 and panels b) and e) of Figure 4 are important new constraints on the sulphate component of the volcanic aerosol, in relation to the hypothesis whether the higher-than-expected strat-AOD could potentially be from non-sulphate aerosol, for example marine aerosol detrained within the volcanic plume.*

To our knowledge, NaCl has been detected once in a stratospheric volcanic plume after the eruption of El Chichon. It was not a marine volcano and produced alkalid magma that could contain halite crystals that were lofted in the plumes (Mechelangeli et al., 1991). In the case of the andesitic Hunga Tonga-Hunga Ha'apai, dissolved marine salt should have been hydrolysed by high temperature water to generate NaOH which reacts with silicium and aluminium to form silicates ($Na_2Si_2O_5$, $NaAlSi_3O_8$, …) in the magma (G. Carazzo, personal communication).Chlorine is released as HCl which is highly soluble. Millan et al, 2022, mention that the HCl perturbation seen by MLS is unexceptional. The presence of halites in the aerosols would haved produced significant depolarization of the lidar signal which is not seen. In the absence of any manifestation of NaCl in the plume and any need to consider this possibility, we are not inclined to take this hypothesis as plausible.

*Readers will be interested then in the specific maximum value of the sulphate AOD shown in these 5 sub-panels of the 2 Figures. In each case however, the legend for the colour plots shown is not apparent currently, with the legend contour-labels shown only for the pale green colouring showing 0.01. Whilst I realise that may well be the intention to enable the reader to infer the aqua colours are above that 0.01 AOD value in Figure 3h), and the same aqua colour has that achieves that AOD>0.01 criteria in Figure 4b, it is not easy to recognise the values for the upper end of the legend contour-scale. I think these are 0.08 in each case, and it would help the reader cross-check these, also with other Figures of the total AOD, in relation to assessing what the measurements are indicating in terms of the proportion of the total AOD that could potentially be non-sulphate.*

*For this reason, please add the 0.08 values to Figure 3h and Figure 4b, or if the value is 0.09 or 0.07 in one case, please set that accordingly.*

*Similarly for Figures 3g, 4a and 4d, please add the upper legend-scale contour label for the value shown for the column SO2 Dobson Unit value shown, and Figure 4f) for the CALIOP integrated scattering.*

The color scale have been homogenized, simplified and properly labelled. See also the corresponding reply to referee 1.

*22) Page 15, Figures 3 and 4 -- THe panel h) of Figure 3 and panels b) and e) of Figure 4*

*The unit used for the integrated scattering is shown as "str" with superscript -1, which I think the authors are using as a shorthand for "per steradian". However, the recognised shorthand is to denote steradian as "sr" rather than "str". I can see that in this specific case, the readers may have chosen to abbreviate the unit as "str" rather than "sr", to avoid confusion with the SR abbreviation for Scattering Ratio.*

*And then I agree that's better in that case, to retain that non-standard three-letter abbreviation of the steradian unit. However, given this non-standard abbreviation, and considering some readers may not be so familar with the "per steradian" unit, I'm requesting the authors give the name of the unit in full in the Figure caption.*

The only two panels where a steradian unit is used are Figs. 2g and 3f, not the one listed here. Nevertheless, we made the suggested modification.

*23) Abbreviation for the sulphate component of the stratospheric AOD in the labels and captions of Figures 3 (panel h) and 4 (panels b and e) -- page 15 -- and in Figure 5 on page 16 and Figure 6 on page 17 (panels a, e and g).*

*The authors have used the abbreviation "SA OD" in the titles of the sub-panel Figures and "SA/OD" in the caption.*

*I have encountered occasional misunderstandings within some communities re: the SAD abbreviation for "Surface Area Density" sometimes being mistakenly stated to be "Sulphate Aerosol Density", similarly to the way the acronym GCM sometimes is sometimes referred to mistakenly as Global Climate Models.*

*I'm further aware that in some published papers the four-letter acronym "SAOD" is used for Stratospheric Aerosol Optical Depth.*

*I am assuming that's why SA is specified sparately there, to aim to ensure readers realise the AOD in this case is actually for the sulphate aerosol component of the optical depth.*

*I'm aware that in some manuscripts (e.g. Dhomse et al., 2014, 2020), a lower-case "s" is used for "stratospheric", which is partly in relation to some readers potentially translating the SA characters within SAOD as sulphate aerosol rather than the shown unit for optical depth of the "stratospheric aerosol".*

*I agree with the point that the abbreviation AOD is the recognised acronym, and prefer then that manuscripts use a lower case s for an sAOD abbreviation for stratospheric AOD.*

*In this case, given that the IMS metric is retrieving a measuremnt of the sulphate component of the aerosol optical depth, it is important that readers correctly interpret the stated acronym.*

*It is important however also to be clear that, at least to my understanding, the optical depth retrieved from the IMS is also that the optical depth is for the sulphate aerosol in the stratosphere.*

*What I am recommending in my review is for the authors to use the abbrevation "SO4" for sulphate, and separate this from the A for aerosol, so then the recognised acronym AOD can be retained within the abbreviation.*

*Given there are two words beginning with S being abbreviated, I'd recommend best to use the lower-case s for sAOD, to abbreviate the retrieved quantity with "SO4-sAOD".*

*If the authors prefer the 4-letter SAOD acronym, this approach also works with "SO4-SAOD", although my preference is for the "SO4-sAOD", on the basis that this is "most easily scanned" or recognised when viewing the Figures.*

*Please use either "SO4-sAOD" or "SO4-SAOD" consistently rather than "SA/OD" or "SA OD", to ensure readers can immediately see it is the sulphate component of the aerosol (in the stratosphere) that the IMS metric is measuring.*

We are a little bit confused about where is the problem and what to do. It should be clear the paper deals with the stratosphere since no data is shown below 18 km, hence it is perhaps not necessary to overload all quantities with a "s" prefix. Then we do not see what it really wrong here with SA/OD.  SA means Sulphate Aerosol which is material entity and OD means Optical Depth which is a physical property. It is then legitimate to separate them by a / or an hyphen. SO4 is a poor descriptor of the sulphates and of the assumed conditions in the IASI/IMS products which are based on the spectroscopic properties of H2SO4 linked with water, which is acceptable as long as water activity is above 1%. SO4 has different spectroscopic properties (Sellitto & Legras, 2016). It is quite common that the same symbol or acronym bears different meanings in sometimes close domains. Besides using Chinese characters or/and adopting the somewhat cryptic notations of biology, we do not see a general solution to this problem that maintains compactness. Within the context of our paper, we do not see where any confusion is possible and SA/OD appears only 6 times in the main text. The figure titles and captions that were using SA OD without / have been put in line with the text.

*24) Please add the wavelength (in subscript) in abbreviations of the AOD*

*Considering the difference in wavelength between the aerosol extinction from OMPS (745nm) and that of the CALIOP lidar (532nm), please add the wavelength*

*Where practical to do so, please include the wavelength either as subscript in the labels for the Figure. Since the right-hand labels are added in Figure 1, this is not essential in that case, but in other Figures it helps cross-comparisons to be aware of the >200nm wavelength difference between the AOD values derived from the OMPS and CALIOP instrument.*

*Related to this issue, also the wavelength of the retrieved stratospheric sulphate AOD should also be given, to ensure then the proportion of the AOD can be interpreted consistently.*

*With changing the acronym to SO4-sAOD, the wavelength can be added with subscript to then ensure the specific measurement of SO4-sAOD can be interpreted to be able to infer the proportion of the stratospheric AOD that is sulphate at the correct wavelength for the IMS retrieval.*

*I must admit I am not sure of the precise wavelength, but am assuming this must be at the mid-visible, at 532nm or 550nm.*

*But please add this wavelength to the caption of the corresponding FIgures, and, given the relevance to the origin of the Hunga Tonga aerosol, please also add this to the labels in the Figures as well. Thanks.*

The wavelength of the backscatter and extinction are mentioned in all figures showing data from CALIOP and OMPS-LP. A few where missing and have been added. Regarding SA/OD, there is here a misunderstanding in spite of the description of IASI/IMS retrieval provided in Appendix A1.2. The SA/OD retrieved from IASI is obtained from an inversion of the infrared spectrum, so it is not associated to a particular wavenumber. Of course the information comes mainly from a few absorbing bands, mainly near 8.5 µm for concentrated sulfuric acid (see Sellitto & Legras, 2016) which are spectrally resolved by IASI and compared to a full spectrum without sulphate. So this is not at all as if the measurement was done by applying a narrow filter at a given wavenumber. It would be therefore very confusing on the origin of the data to overload the IMS SA/OD product with a given wavenumber, certainly not in the visible range anyway, and we cannot see any example of such usage among IASI products in the literature.

References

- Brenna, M., Cronin, S. J., Smith, I. E. M., Pontesilli, A., Tost, M., Barker, S., Tonga'onevai, S., Kula, T., and Vaiomounga, R.: Post-caldera volcanism reveals shallow priming of an intra-ocean arc andesitic caldera: Hunga volcano, Tonga, SW Pacific, Lithos, 412–413, 106614, *https://doi.org/10.1016/j.lithos.2022.106614*, 2022.
- Chouza, F., Leblanc, T., Barnes, J., Brewer, M., Wang, P., and Koon, D.: Long-term (1999–2019) variability of stratospheric aerosol over Mauna Loa, Hawaii, as seen by two co-located lidars and satellite measurements, Atmos. Chem. Phys., 20, 6821–6839, https://doi.org/10.5194/acp-20-6821-2020, 2020.
- Gorkavyi, N., Krotkov, N., Li, C., Lait, L., Colarco, P., Carn, S., DeLand, M., Newman, P., Schoeberl, M., Taha, G., Torres, O., Vasilkov, A., and Joiner, J.: Tracking aerosols and SO2 clouds from the Raikoke eruption: 3D view from satellite observations, Atmos. Meas. Tech., 14, 7545–7563, https://doi.org/10.5194/amt-14-7545-2021, 2021.
- Khaykin, S., Legras, B., Bucci, S., Sellitto, P., Isaksen, L., Tencé, F., Bekki, S., Bourassa, A., Rieger, L., Zawada, D., Jumelet, J., and Godin-Beekmann, S.: The 2019/20 Australian wildfires generated a persistent smoke-charged vortex rising up to 35 km altitude, Commun Earth Environ, 1, 22, https://doi.org/10.1038/s43247-020-00022-5, 2020.

- Khaykin, S., Podglajen, A., Ploeger, F., Grooß, J.-U., Tence, F., Bekki, S., Khlopenkov, K., Bedka, K., Rieger, L., Baron, A., Godin-Beekmann, S., Legras, B., Sellitto, P., Sakai, T., Barnes, J., Uchino, O., Morino, I., Nagai, T., Wing, R., Baumgarten, G., Gerding, M., Duflot, V., Payen, G., Jumelet, J., Querel, R., Liley, B., Bourassa, A., Hauchecorne, A., Ravetta, F., Clouser, B., and Feofilov, A.: Global perturbation of stratospheric water and aerosol burden by Hunga eruption, https://doi.org/10.1002/essoar.10511923.1, 20 July 2022.

- Lestrelin, H., Legras, B., Podglajen, A., and Salihoglu, M.: Smoke-charged vortices in the stratosphere generated by wildfires and their behaviour in both hemispheres: comparing Australia 2020 to Canada 2017, Atmos. Chem. Phys., 21, 7113–7134, https://doi.org/10.5194/acp-21-7113-2021, 2021.

- Michelangeli, D. V., Allen, M., and Yung, Yuk. L.: Heterogeneous reactions with NaCl in the El Chichon volcanic aerosols, Geophys. Res. Lett., 18, 673–676, https://doi.org/10.1029/91GL00547, 1991.

- Millán, L., Santee, M. L., Lambert, A., Livesey, N. J., Werner, F., Schwartz, M. J., Pumphrey, H. C., Manney, G. L., Wang, Y., Su, H., Wu, L., Read, W. G., and Froidevaux, L.: The Hunga Tonga-Hunga Ha'apai Hydration of the Stratosphere, Geophysical Research Letters, 49, e2022GL099381, https://doi.org/10.1029/2022GL099381, 2022.

- Poli, P. and Shapiro, N. M.: Rapid Characterization of Large Volcanic Eruptions: Measuring the Impulse of the Hunga Tonga Ha'apai Explosion From Teleseismic Waves, Geophysical Research Letters, 49, https://doi.org/10.1029/2022GL098123, 2022.

- Schoeberl, M., Ueyama, R., Taha, G., Jensen, E., and Yu, W.: Analysis and impact of the Hunga Tonga-Hunga Ha'apai Stratospheric Water Vapor Plume, https://doi.org/10.1002/essoar.10511762.1, 5 July 2022.

- Sellitto, P. and Legras, B.: Sensitivity of thermal infrared nadir instruments to the chemical and microphysical properties of UTLS secondary sulfate aerosols, Atmospheric Measurement Techniques, 9, 115–132, https://doi.org/10.5194/amt-9-115-2016, 2016.

- Sellitto, P., Podglajen, A., Belhadji, R., Boichu, M., Carboni, E., Cuesta, J., Duchamp, C., Kloss, C., Siddans, R., Begue, N., Blarel, L., Jegou, F., Khaykin, S., Renard, J.-B., and Legras, B.: The unexpected radiative impact of the Hunga Tonga eruption of January 15th, 2022, https://doi.org/10.21203/rs.3.rs-1562573/v1, 18 April 2022.

- Taha, G., Loughman, R., Colarco, P. R., Zhu, T., Thomason, L. W., and Jaross, G.: Tracking the 2022 Hunga Tonga-Hunga Ha'apai aerosol cloud in the upper and middle stratosphere using space-based observations, 2022.

- Wright, C. J., Hindley, N. P., Alexander, M. J., Barlow, M., Hoffmann, L., Mitchell, C. N., Prata, F., Bouillon, M., Carstens, J., Clerbaux, C., Osprey, S. M., Powell, N., Randall, C. E., and Yue, J.: Surface-to-space atmospheric waves from Hunga Tonga-Hunga Ha'apai eruption, Nature, https://doi.org/10.1038/s41586-022-05012-5, 2022.

- Zhu, Y., Bardeen, C., Tilmes, S., Mills, M., Harvey, V., Taha, G., Kinnison, D., Yu, P., Rosenlof, K., Wang, X., Avery, M., Kloss, C., Li, C., Glanville, A., Millán, L., Deshler, T., Portmann, R., Krotkov, N., and Toon, O.: 2022 Hunga-Tonga eruption: stratospheric aerosol evolution in a water-rich plume, https://doi.org/10.21203/rs.3.rs-1647643/v1, 12 May 2022.

---

## Author Response (AR2)

ANSWER TO THE EDITOR

We thank the editor for her prompt answer and we provide below a detailed answer to her comments. The difference file collates the difference file with respect to the submission and the difference file with respect to the first revision. The two separate files are also available from the following link

https://filesender.renater.fr/?s=download&token=dd19bafa-93d8-4f3c-ad06-c8f07c047291

I really appreciate the effort you have put in the revision of your manuscript and in keeping the word limits despite all the additional information requested by the referees.

However, now the first part of the manuscript is difficult to read and hard to follow. I am wondering if you are somewhat too ambitious with trying to discuss and describe six figures in a short letter paper. I guess that this was also the reason why referee 1 thought this is supposed to be a regular manuscript.

We agree, for instance, that the sentences near P3L35 are somewhat difficult to read and do not make a good start for the discussion but this is due to your request to expand the names of the three satellite instruments made in your first report. This does not need to occur in standard ACP format as the data and method section comes first. The problem arises here because the data and method section has been transferred in the appendix. The practice of other journals with such format is to allow also the full definition to be transferred to the appendix when there are many instruments involved (e.g., see Khaykin et al., 2020). It is not also that MLS, CALIOP and OMPS-LP are very new instruments which have never been heard of in ACP. In the absence of a clear policy for ACP Letters and to improve clarity, we remove the expanded acronyms from the main text.

We maintain the full spelling in the text for ALADIN and IMS as this is, for the first, a fairly recent and not so well-known instrument and, for the second, a new product used only in a few papers so far.

We have also tried to improve other parts of the text where you found necessary to clarify and some where we found this need by ourselves.

Therefore, I suggest another round of major revisions. I am really sorry for that, but in the end this will be a nice, clear paper that is useful for the scientific community. Therefore, this changes are worth the effort.

First of all you should rethink your choice of manuscript type. If you still prefer the letter style, you probably should skip the detailed discussion of some figures in the main text and put these rather in the appendix and provide there the necessary information. Otherwise, if you still have difficulties keeping the word limits with the content you would like to provide, then you should extend and rewrite your paper to a regular ACP paper. It then still can be a highlight paper.

The appendix is used to describe the methods, following the recommendations for the preparation of ACP Letters manuscript https://www.atmospheric-chemistry-and-physics.net/about/manuscript_types/acp_letters.html and we done thing it is where the discussion of results should take place.

As the consequence, the two figures that were in the appendix are now numbered with the others, thus satisfying another requirement of the editor.

Mainly revision for sections 2-4 are necessary. The discussion and conclusion were fine. Here are my detailed comments:

General: To my knowledge the Stratospheric Aerosol Optical Depth is abbreviated SAOD without a slash in between. I would thus suggest to remove the slash.

SA/OD is for Sulphate Aerosol / Optical Depth. Other aerosol types are not retrieved with this product which is based on the specific spectral properties of sulphate aerosols (H2SO4, water and sulphur-containing ions in sulphate aerosols droplets). The second referee explains at length why it would be confusing to use SAOD. He unfortunately makes a suggestion which would be equally or even more confusing (to call this "SO4-AOD"). The product is new and has been produced by two of the co-authors of the paper who might have a say on how it is named. After discussion with them, and in order to remove any further source of discussion, it has been decided to rename it as "SA optical depth" in the text and "SA OD" with a separating blank in the figures. SA is the general acronym used for "Sulphate Aerosol" in many publications.

Title: The length is definitely ok, but I cannot understand the second part. What do you mean with "vertical separation from the water vapour"?

We prefer the short initial version but this was added at the request of the second reviewer. Both reviewers wanted to expand the title and we followed the suggestion of the second one who is apparently a native English speaker. We are not very happy of that since the new title focuses on only one of our main results. If you make such a recommendation, we would be inclined to return to the initial version but since, after all, our paper is mainly about the layer of sulphate aerosols, and there are other papers published or in review that are more focused on the water plume, we can take the suggestion of the first referee and  the title is now "The evolution and dynamics of the Hunga Tonga-Hunga Ha'apai sulphate aerosol plume in the stratosphere"

P1, L4: You mean "sulphate aerosols"? Then you should also add "aerosol".

Done

P1, L8: hydration is here not the correct term. You mean condensation? If the particles take up the water vapour it is condensation, if the water vapour in the atmosphere is enriched due to the water containing aerosol particles then you may talk about hydration. You have to be definitely clearer when you use these terms.

We find that condensation alone would be ambiguous here as it is not clear which species condensates. Hydration is indeed a shortcut for hygroscopic growth, water uptake or water condensation. Hygroscopic growth is the most accurate as this notion is usually associated with the thermodynamical equilibrium. Hence, for the sake of clarity, we modify hydration to hygroscopic growth

P1, L8-9: ".….and separate from the equally persistent and ascending moisture in the Brewer-Dobson circulation." Also here the different processes are not clear and the discussion rather confusing. When you have moisture entrained then this is related to the upward motion within the Brewer-Dobson circulation, but the sedimentation is the gravitational settling of aerosol particles, thus a downward motion.

We changed "As sulphate particles grew through hydration and coagulation, they sediment and separate from the ascending moisture entrained in the Brewer-Dobson circulation" to "Sulphate particles, undergoing hygroscopic growth and coagulation, sediment and gradually separate from the moisture anomaly entrained in the ascending branch Brewer-Dobson circulation."

It occurs to us that the editor is not unfamiliar with the idea that sedimentation of particles in the stratosphere leads to a vertical separation of the compounds which then follow a different evolution (e;g. Khosrawi et al., 2011, https://doi.org/10.5194/acp-11-8471-2011)

P1, L10-11: I would suggest to combine the last two sentence, so that it reads: "………….with vorticity anomalies that may have enhanced the duration and impacts of the plume".
General on the abstract: In order to have a clear, concise abstract I would suggest that you define three key points reflecting your results first and then write from this your abstract.
P1, L13: of -> on

P1, L17: space before the comma obsolete

P2, L22: add missing reference

All done

P2, L26: The major point is that you "advocate that its climatic effect is very significant": I would suggest to rephrase the sentence so that this becomes clearer (try to put this at the end of the sentence).

Done

P2, L28: It is really weird that you start your paper with discussing an appendix figure. If this figure is important it should belong to the main text.

We agree it is somewhat unusual to start by referring to an appendix figure. Our general choice was to have only figures showing instrumental results in the main text. This figure is ancillary but nevertheless important. It has been moved as Fig. 1 without any other change, than shifting the figure numbers. Fig. A2 has followed as it makes no sense to keep it isolated.

P2, L30: add "at" so that it reads "except at a narrow…."

Done

P2, L31: rotation? Do you mean rotation speed? Then you should also write "rotation speed".

Done.

P2, L32: effects -> effect

P2, L34: add "latitude" so that it reads "latitude band".

P2, L36: one comma obsolete and delete "below". Rather than the section you should refer to the respective figure.

P2, L37: a mid-April maximum -> a maximum in mid-April

P2, L43: delete "calculated following" and just give the appendix in parenthesis, thus write "(Appendix 3)".

All done

P2, L44: "……using Eq 9.42 from Seinfeld and Pandis (2016)" put this information into the appendix

Unfortunately, it does not fit in the Appendix A3 and we cannot reasonably make an appendix just for that. We can safely remove this information from the text as it is also given in the caption of Fig. 3 (now Fig. 4).

P3, L50: "…… ERA5 upwelling"……not clear, rephrase.

We have added "the" in front. The word upwelling is common in papers about the stratospheric circulation.

P3, L53: aerosol size after growing up to -> aerosol particle size growing to

Done

P3, L54-56: Rewrite as follows (make one sentence): The observed trend of the extinction-to-backscatter ratio obtained from OMPS-LP and CALIOP (Fig. 2f) is consistent with the theoretical trend direction for sizes between 1 and 2 mu m." What are the theoretical trend directions? Is a reference needed here? The information on the calculation using the Mie code can be provided in the appendix or add after bascatter ratio in parenthesis obtainted also from the Mie code. However, the "also" is quite confusing here. You haven't mentioned any Mie code calculation before in the text.

The three sentences have been rewritten as "The extinction-to-backscatter ratio, obtained by combining OMPS-LP and CALIOP data (Fig. 3g) exhibits a growth followed by a decay which are qualitatively consistent with the aerosol equivalent sedimentation speed evolution (Fig. 3e) and the expected behavior of the ratio (Fig.3f) in the 1-2 μm size range." Previous Fig.2 is now Fig.3.

P3, L57: something missing here? The textpart "and the decoupling between aerosols and moisture" makes no sense.

We have changed the word "decoupling" by "progressive vertical separation". All the previous paragraphs are about the opposite vertical motion of aerosols and water. Notice that the sedimentation alone is responsible for the separation. It does not matter whether the ambient air is rising, as it is here, or descending like in the winter polar vortex (as in Khoswari et al., 2011).

P3, L58: hydration -> here you mean condensation. Particles grow by condensation.

We use hygroscopic growth, see above

P3, L58: skip "growth" and just write "coagulation" or write "growth by coagulation"

Coagulation alone saves two words, so this is the choice.

P3, L63: such -> these

Done

The two lines P3L64-66 have been discarded as this remark does not contribute to our conclusions.

P3, L70: within -> of

P3, L74: brings confirmation -> confirms this

P3, L78: suggesting -> suggesting that

P4, L91: add " ratios" so that it reads "depolarization ratios".

All done

P4, L94: skip here the text in parenthesis and add this rather at line 96 where you mention again the movie.

We prefer to keep it with no change. This was specifically added here under the request of referee 1 who wanted to be sure that Fig.5 (now Fig.6) alone is able to support what follows and that the movie is only provided an extended view)

P4, L99: Be clearer here. An early case of what?

Of concentrated patches. We think it is clear from the two previous sentences.

P4, L104: What do you mean with section? The track?

A capital was missing, we mean Section

P4, L105: we? You mean CALIOP? Then write CALIOP.

We the authors and the reader.

P4, L108: in SAOD -> in the SAOD

P4, L109: supplement -> supplemental

P5, L117: suggest -> suggest that

All done

P5, L123: "normal" obsolete?

 We replaced "normal" with "background"

P5, L129: as shown from -> as seen in the

"as seen from" since the raw measurements need to be processed to make the result apparent

P5, L130: water vapour -> water vapour distribution (?)

P5, L133: hydration -> condensation

P5, L143: evaporation causes a shrinking rather than a growing and hydration should read condensation.

We use "hygroscopic growth". See above

P8, L217: skip "the" before stratospheric

P8, L217: are pure -> are on pressure

Done

---

## Author Response (AR3)

Answer to the comments of referee 1.

Some minor comments:
L6: change to Still persist after six months
L30: Is the "and 5S" needed here. I don't really understand it please clarify
The authors could add in "(Figure1a)"
Done as suggested

L44: This sentence is really confusing.
"Figure 3b and d-e show a fast descent is visible in the two latitude bands (Fig. 3a-d) which would imply unrealistically large aerosol sizes (Fig. 3e)."
Why are the authors referring to b and d at the beginning of the sentence? only b and d show vertical montion. Why are then referring to Figure 3a-d.
This sentence was really confusing with the figure references, probably a consequence of a wrong cut and paste. It has been fixed.

The section extinction of the plume could be improved if the authors were to mark the clouds, for example, C1 and C2 in the figure.
Then for example in line 82, the authors could say "Comparing Figs. 5a and b, makes apparent that the conversion to sulphates is almost complete in the western strip associated to C1 in Fig. 4e while it is incomplete in the eastern strip associated with C2." And if the authors add these labels to these subplots everything will be much easier.
L102: If the authors use the C1 and C2 labels, this will be much clearer
We have applied this suggestion which indeed simplify the wording and makes easier to follow the discussion.

L124: the remaining plume -> remaining sulphate plume
No, at this stage, the plume contains (at least) SO2, sulphate and water.

L132 Do the authors mean 2 and 3
Yes, corrected.

L139 Do the authors mean 1.4 as discussed through the rest of the paper
Yes, corrected.

L145. There is no reason to assume this, nor is needed for the rest of the article.
We think it is a reasonable assumption. This comment is independent of the rest of the article but we also think it is a useful element of the discussion. Our findings support a fast conversion sulphate that suggests prior estimates of SO2 injection that ignore this fact are too low.

L176 only SO2 or you meant to say and volcanic-specific trace gases (SO2) Can you not do that with RTTOV 11?
Please clarify
RTTOV 11 does resolve the SO2/sulphate bands and cannot be used in this retrieval.

L195 please add: as well as other trace gases, temperature and cloud ice.
L196 .. grid of 1.45 (add space) resolution
Done.

L202: This only affects 1d retrievals. From Gorkavyi et al 2021 "One way to account for such effects is to use a two-dimensional (2D) radiative transfer model (RTM) that is able to account for such effects along with multiple observations in a tomographic retrieval (e.g., Livesey et al., 2006; Zawada et al., 2018; Loughman et al., 2018)." The first citation is the manuscript that describes the MLS retrieval, that is, the MLS products should not be affected by the arch effect. Please move this discussion of the arch effect to the OMPS section.
Thanks for this. The sentence has been moved to the OMPS section. It seems, however, that during the early phase, the bottom boundary of the MLS water vapour anomaly is in better agreement with OMPS than CALIOP.

L212 This statement needs a citation
A citation has been added.

L270 The link is not working

This temporary institutional link has been replaced by a link to the zenodo depository with a doi.

We do wish to apply the changes suggested by referee 2. His lengthy report repeats comments that he made previously and that we already answered. He is basically wishing us to alter our analysis and conclusions to mention that ultra-thin ash could have been hidden within sulfate aerosol droplet. This is a possibility upon which one can speculate but there is nothing in the available data that detect such compounds in the two main components of the plume released initially at 31 and 28 km that are the focus of our study. The referee wrongly claims that he sees a strong depolarization in our figure 4f where it is shown that it does not exceed 1%. There is nobody having some experience with lidar data who would not say that this is very small depolarization. And this magnitude is preserved for the dispersed plume over several months until it is not possible to assess the depolarization. In comparison, the thin cloud seen on 15 January by CALIOP at 35 km, mentioned by referee 2 and that we mentioned ourselves on the SSIRC VOLC exchange forum, was depolarizing 50%. This is large depolarization, not 1%. This thin cloud was seen also on 16 January by CALIOP and on 20 January over La Reunion and then was lost. We did not mention this event because we consider it as an epiphenomenon related to the two main clouds which generated a persistent plume and because it is mentioned in Khaykin et al., 2022, which shares a number of common authors. The referee further speculates that if CALIOP was not turned off during one week between 19 and 26 January because of solar activity we would have seen ash aerosols. This is a very imaginative analysis of unrecorded data. However, Baron et al. (2022) report lidar measurements at La Reunion during that week and Taha et al. (2022) report TROPOMI UV measurements which both do not see ash. Therefore we tend to stick to our interpretation that the ash was removed by the ice condensation and fall that followed the huge initial injection of water, leaving a saturated stratosphere up to 35 km (see Sellitto et al. and Khaykin et al, 2022). The remaining ash, if any, cannot be optically detected in the main plume and therefore can be safely ignored. We have added a sentence in the conclusion mentioning that the absence of detection does not rule out the possibility of hidden presence but we cannot go further. The referee 2 makes a few other suggestions for rewording that we consider as purely obstructive as this stage.